# TimeSliver : Symbolic-Linear Decomposition for Explainable Time Series Classification

**Akash Pandey**[*]  **Payal Mohapatra**[*]  **Wei Chen**  **Qi Zhu**[†]  **Sinan Keten**[†]
Northwestern University, Evanston, IL, USA

## Abstract

Identifying the extent to which every temporal segment influences a model's predictions is essential for explaining model decisions and increasing transparency. While post-hoc explainable methods based on gradients and feature-based attributions have been popular, they suffer from reference state sensitivity and struggle to generalize across time-series datasets, as they treat time points independently and ignore sequential dependencies. Another perspective on explainable time-series classification is through interpretable components of the model, for instance, leveraging self-attention mechanisms to estimate temporal attribution; however, recent findings indicate that these attention weights often fail to provide faithful measures of temporal importance. In this work, we advance this perspective and present a novel explainability-driven deep learning framework, TimeSliver, which jointly utilizes raw time-series data and its symbolic abstraction to construct a representation that maintains the original temporal structure. Each element in this representation linearly encodes the contribution of each temporal segment to the final prediction, allowing us to assign a meaningful importance score to every time point. For time-series classification, TimeSliver outperforms other temporal attribution methods by 11% on 7 distinct synthetic and real-world multivariate time-series datasets. TimeSliver also achieves predictive performance within 2% of state-of-the-art baselines across 26 UEA benchmark datasets, positioning it as a strong and explainable framework for general time-series classification.

## 1 Introduction

Deep-learning (DL) models such as Convolutional Neural Network (CNN), Long Short-Term Memory (LSTM), and Transformer have proven to be successful as predictive models for time series classification tasks. However, while most DL models offer strong predictive performance, they are not interpretable, limiting our understanding of their decision-making process (Rudin, 2019; Doshi-Velez & Kim, 2017). Interpretable DL models are essential for trust and transparency, particularly in high-stakes domains such as healthcare, law, and finance, where explanations support informed decision-making and regulatory compliance (Rudin, 2019). They also help detect biases in training data, ensure fairer outcomes (Caruana et al., 2015; Molnar, 2020), and facilitate the extraction of new scientific knowledge (Pandey et al., 2025).

Over the past few years, several methods have been developed to explain the decisions of DL models. Popular methods like DeepLift and Integrated Gradients attribute predictions via baseline-based backpropagation or path integrals but require careful baseline selection (Shrikumar et al., 2017; Sundararajan et al., 2017). Another method called Grad-CAM, a purely gradient-based approach, attributes importance via output–feature derivatives, but is tailored for CNNs and performs poorly on temporal attribution tasks (Selvaraju et al., 2017; Saha et al., 2024). SHAP-based approaches leverage game-theory-based Shapley scores to provide consistent explanations via a unified framework, but they assume feature independence and scale poorly with dimensionality (Lundberg & Lee, 2017). All these **post-hoc interpretability methods** face shared challenges of high parametric sensitivity and explanations that often vary significantly across datasets (Turbé et al., 2023).

Another set of approaches advocates **explainability based on the model's inherent components**. For instance, some works leverage self-attention weights (Wu et al., 2020; Clark et al., 2019; Rogers et al.,

---

[*]Equal contribution
[†]Equal advising

2021; Vig et al., 2021) in Transformers (Vaswani et al., 2017) as a key tool for model explainability. Grad-SAM (Barkan et al., 2021) enhances this by weighting attention scores with their output gradients. However, due to the non-linearities in Transformers, attention weights often fail to align (unfaithful) with ground-truth attribution (Chefer et al., 2021; Serrano & Smith, 2019; Jain & Wallace, 2019; Wiegreffe & Pinter, 2019). Another instance is a recent Multiple Instance Learning (MIL)-based temporal attribution method (Early et al., 2024), which shows promising results in identifying the importance of each time point. However, it has not been extended to multivariate time-series settings and has limited experimental comparisons only to Grad-CAM and DeepLiftSHAP. Some more recent approaches use self-supervised model behavior consistency (Queen et al., 2023) or the modified information bottleneck (Liu et al., 2024) to compute attribution scores, but they either depend on a pretrained model (Queen et al., 2023) or are computationally complex due to multiple components and hyperparameters (Liu et al., 2024). In protein modeling, COLOR (Pandey et al., 2025) enhances explainability by segmenting protein sequences into motifs Pandey et al. (2024) for representation learning. However, it cannot differentiate between positively and negatively attributing segments, and protein sequences are inherently univariate and composed of categorical variables, unlike multivariate continuous time series data. These limitations *(non-linearities leading to unfaithful attributions, inapplicability to multivariate time series, sensitivity to hyperparameters)* of prior methods motivate us to explore an **explainability-driven predictive modeling approach capable of handling multivariate time series** with robust attribution capabilities across domains.

In this work, we introduce `TimeSliver`, a novel deep learning model that computes Temporal attribution using Symbolic–Linear Vector Encoding for Representation. `TimeSliver` processes raw (uni- or multi-variate) time series and their symbolic counterparts (via binning) to produce localized, segment-level representations. These representations are then linearly combined into a sequence-length-independent, explainable representation that enables the computation of temporal attribution scores and facilitates insight into the model's predictions. Our main contributions are as follows:

- ▸ We propose an **explainability-driven deep learning framework**, `TimeSliver`, which learns compact representations through a novel linear composition of symbolic and latent representations of temporal segments to provide temporal importance for multivariate Time Series Classification (TSC) tasks while maintaining state-of-the-art predictive capacity (Section 2.2.3).

- ▸ `TimeSliver` provides **positive and negative temporal attribution scores** to offer a complete explanation of different time points for the model's prediction (Section 2.2.4).

- ▸ We evaluate `TimeSliver`'s explainability across **three diverse real-world applications**—audio, sleep-stage classification, and machine fault diagnosis—a**s well as on four synthetic datasets, against twelve baseline methods**, which place `TimeSliver` consistently as a **top-performing model for identifying positively and negatively influencing temporal segments** under various settings (Section 3.1).

- ▸ We demonstrate `TimeSliver`'s **competitive predictive performance on 26 multivariate time-series classification tasks** from the UEA benchmark (Section 3.2).

**Additional Related Works.** Decomposing time-series inputs into *human-understandable patterns* also contributes to explainability. Recent works explore approaches such as shapelet decomposition (Wen et al., 2025b), reinforcement learning-based subsequence selection (Gao et al., 2022a), and abstracted shape representations (Wen et al., 2024). In particular, learnable shapelet-based methods (Wen et al., 2025b; Li et al., 2021a; Qu et al., 2024a; Ma et al., 2020) for encoding subsequences are popular pattern-based explainable models. These methods are generally better suited for qualitative assessment and exhibit varied performance metrics, making them challenging to benchmark (Wen et al., 2024; 2025b). Another class of *self-explainable* methods uses neuro-symbolic approaches (Yan et al., 2022) with signal temporal logic (Mehdipour et al., 2020) to output soft-logic predicates at each time step. Architecturally, explainability can also be incorporated through concept bottleneck networks (CBMs) (Koh et al., 2020), which introduce *human-understandable concepts* as intermediate predictions. However, CBMs typically require dense concept annotations and manual editing, practices often impractical in high-stakes applications. Some recent works address this by proposing data-efficient CBMs (Koh et al., 2020) and exploring their applicability in time-series settings (van Sprang et al., 2024; Wen et al., 2025b).

## 2 METHODOLOGY

### 2.1 PRELIMINARIES AND NOTATIONS

Although DL models such as 1D CNNs, LSTMs, and Transformers have proven effective for time-series prediction, they often lack explainability, particularly in terms of *temporal attribution*.

**Definition 2.1** (Temporal Attribution-Based Explainability). Temporal attribution-based explainability in time series refers to the process of assigning importance scores to each time point in an input sequence by decomposing them into positive and negative contributions. Positive attribution scores quantify how much each time point drives the prediction toward the predicted class, while negative attribution scores quantify how much each time point drives the prediction away from the predicted class. Together, these scores enable identification of which time steps most significantly influence the model's prediction.

**Problem Statement.** Given a dataset $\mathcal{D} = \{(\mathbf{x}_i, y_i)\}_{i=1}^{N}$ with $N$ samples, where $\mathbf{x}_i \in \mathbb{R}^{L \times v}$ is a multivariate time-series input of length $L$ with $v$ features (or number of input channels), and $y_i \in \{0, \ldots, C-1\}$ is the corresponding class label, we aim to learn an explainable model $f$ by learning a latent representation using parameters $\theta_\mathrm{q}$ and then projecting it using a linear layer with parameters $\theta_\mathrm{c}$ to predict output logits $\hat{y}_i \in \mathbb{R}^C$, given as

$$\hat{y}_i = f(\mathbf{x}_i; (\theta_\mathrm{q}, \theta_\mathrm{c})) \in \mathbb{R}^C.$$

Based on the optimized parameters $(\theta_\mathrm{q}^*, \theta_\mathrm{c}^*)$, we assign positive and negative temporal attribution scores for each time point $k \in [1, L]$ in the input sample $\mathbf{x}_i$, given as

$$\{\phi_k^{+(i)}, \phi_k^{-(i)}\}_{k=1}^{L} = f_\mathrm{att}\big(\mathbf{x}_i, (\theta_\mathrm{q}^*, \theta_\mathrm{c}^*), \hat{y}_i\big).$$

The attribution function $f_\mathrm{att}$ is derived from the internal representations of the input $\mathbf{x}_i$ and the outputs of $f$.

### 2.2 OUR APPROACH

In this section, we present our explainability-driven novel deep learning model, `TimeSliver`, illustrated in Figure 1 which comprises of three key modules: **(I)** conversion of the raw time-series input $\mathbf{x}_i$ into temporal segments and learning their representations $\boldsymbol{Q}$, **(II)** construction of a latent temporal vector $\boldsymbol{Z}$ using symbolic abstraction of $\mathbf{x}_i$, and **(III)** a linear composition of $\boldsymbol{Q}$ and $\boldsymbol{Z}$ to yield a representation of $\mathbf{x}_i$ that maintains initial temporal structure. This combined representation is fed to a *linear layer*, denoted as $f_\mathrm{cls}$ component, to predict the target label $y_i$. Using the input representations, $\boldsymbol{Q}$, $\boldsymbol{Z}$ and the logits from $f_\mathrm{cls}$, we compute temporal attribution scores via the non-parametric operation $f_\mathrm{att}$ described in Section 2.2.4.

**Definition 2.2** (Temporal Segment). Given a multivariate time series instance $\mathbf{x}_i \in \mathbb{R}^{L \times v}$, a *temporal segment* is defined as a contiguous sub-sequence of $\mathbf{x}_i$ of length $m$ (with $m \leq L$). Formally, a segment is $\mathbf{x}_s = \mathbf{x}_i[t : t + m] \in \mathbb{R}^{m \times v}$, where $t$ is its start index in $\mathbf{x}_i$

#### 2.2.1 MODULE I: LATENT REPRESENTATION OF TEMPORAL SEGMENTS

Given a multivariate time-series input $\mathbf{x}_i \in \mathbb{R}^{L \times v}$, this module converts $\mathbf{x}_i$ into $\kappa = L - m + 1$ overlapping latent representations. These representations are obtained using a 1D convolutional operator with kernel size $m$ and stride 1 resulting in $\kappa$ latent feature vectors in a sequence. Each latent feature vector captures a localized temporal context within the time series. The 1D CNN is parameterized by learnable weights $\boldsymbol{\theta_q}$ and transforms each segment into a $q$-dimensional latent representation, resulting in a matrix $\boldsymbol{Q} \in \mathbb{R}^{\kappa \times q}$. Formally, this is defined by a learnable mapping, $\mathbf{g}(\mathbf{x}_i; \theta_\mathrm{q}) \mapsto \boldsymbol{Q} = [\mathbf{q}_1; \mathbf{q}_2; \ldots; \mathbf{q}_\kappa]^T$, where $\mathbf{q}_j \in \mathbb{R}^q$ is the latent vector for the $j^\mathrm{th}$ segment, enabling end-to-end learning of localized temporal patterns.

#### 2.2.2 MODULE II: SYMBOLIC COMPOSITION-BASED REPRESENTATION

In this module, each variate $\mathbf{x}_i^{(j)} \in \mathbb{R}^L$, for $j \in \{0, \ldots, v-1\}$, is independently discretized into one of $n$ categorical bins using a fixed binning strategy, as proposed by Lin et al. (2007).

Code is available at https://github.com/pandeyakash23/TimeSliver

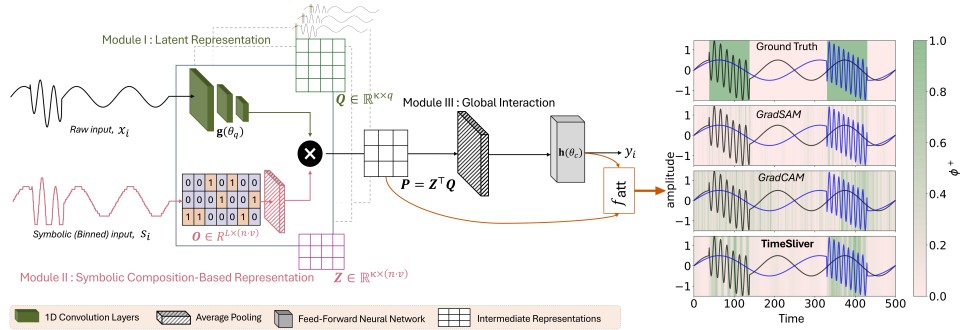

Figure 1: Overview of `TimeSliver`: (Module I) temporal segment extraction and latent representation learning $(\mathbf{g}(\mathbf{x}_i; \theta_q))$; (Module II) symbolic composition of temporal segments; and (Module III) global **linear** interaction between latent and symbolic representations to generate $\boldsymbol{P}$, a representation of $\mathbf{x}_i$ preserving temporal structure. $\boldsymbol{P}$ is then passed through a **linear layer** $(\mathbf{h}(\mathbf{x}_i; \theta_c))$ to predict $y_i$ and used to compute temporal attribution using $f_{att}$. The right column compares ground truth attribution scores with baseline methods and `TimeSliver`, where darker regions indicate positive influence.

This yields a symbolic matrix $s_i \in \{1, \ldots, n\}^{L \times v}$, where each element $s_i^{(t,j)}$ indicates the symbolic bin index assigned to the $j^{\text{th}}$ variate at time step $t$. The symbolic representation is formally defined as $s_i = h(\mathbf{x}_i; n, w)$, where $h(\cdot)$ is a deterministic discretization function parameterized by the number of bins $n$ and the compression window size $w$. In this work, we choose $w = 1$.

Next, we convert $s_i$ into a one-hot encoded matrix $\mathcal{O} \in \mathbb{R}^{L \times (n \cdot v)}$ by independently applying one-hot encoding to each variate and concatenating the results along the feature dimension. Specifically, for each variate $j \in \{0, \ldots, v-1\}$, we construct a one-hot matrix $\mathcal{O}^{(j)} \in \{0,1\}^{L \times n}$, where the $t^{\text{th}}$ row $\mathcal{O}_t^{(j)}$ corresponds to the one-hot encoding of the symbolic value $s_i^{(t,j)}$. The final matrix is formed as:

$$\mathcal{O} = \left[ \mathcal{O}^{(1)} \| \mathcal{O}^{(2)} \| \cdots \| \mathcal{O}^{(v)} \right] \in \mathbb{R}^{L \times (n \cdot v)},$$

where $\|$ denotes concatenation along the column (feature) axis. In alignment with past works (Esmael et al., 2012; Combettes et al., 2024; Mohapatra et al., 2025a) noting that using a shared symbolic embedding space across variates can lead to semantic ambiguity and information loss, this structured symbolic encoding ensures that each variate-specific semantic identity is retained.

To obtain a segment-wise symbolic representation aligned with the temporal segments extracted in Section 2.2.1, we apply average pooling over the one-hot encoded matrix $\mathcal{O} \in \mathbb{R}^{L \times (n \cdot v)}$ using a sliding window of size $m$ and stride 1. This yields a symbolic composition matrix $\boldsymbol{Z} \in \mathbb{R}^{\kappa \times (n \cdot v)}$ as shown in Figure 2b, where each entry $Z_{ij}$ captures the normalized frequency of the $j^{\text{th}}$ symbolic feature within the $i^{\text{th}}$ segment as shown in Figure 2c. Formally, this is computed as:

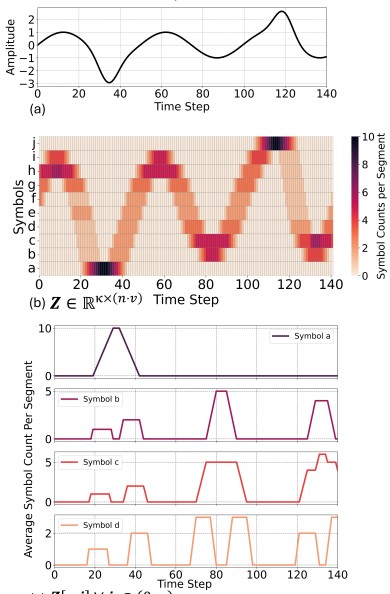

Figure 2: (a) shows a raw time series input, (b) is the symbolic composition matrix, $\boldsymbol{Z}$ and (c) shows some sample rows of $\boldsymbol{Z}$ which serve as the **Bag-of-Stencils** to modulate $\boldsymbol{P}$.

$$Z_{ij} = \frac{1}{m} \sum_{l=0}^{m-1} \mathcal{O}_{i+l, j}, \quad \text{for } i \in \{0, \ldots, \kappa-1\}, \ j \in \{0, \ldots, n \cdot v - 1\}, \tag{1}$$

where $\mathcal{O}_{t,j}$ denotes the $j^{\text{th}}$ one-hot dimension at time step $t$. Thus, each row $\boldsymbol{Z}_{i:}$ represents the symbolic distribution over the $i^{\text{th}}$ temporal segment.

**Remark (Approximate Structural Analogy of $\boldsymbol{Z}$ with Spectral Representation).** The Short-Time Fourier Transform (STFT) (Oppenheim et al., 1999) for a segment $i$ (of length $m$) for an input $x$ computes the energy at frequency $f$ as,

$$S_{if} = \frac{1}{m} \left| \sum_{l=0}^{m-1} x[i+l] \, e^{-j2\pi f l/m} \right|^2$$

providing a localized decomposition of $x$ onto the orthonormal sinusoidal basis $\{e^{-j2\pi fl/m}\}_{f=0}^{n-1}$, with $S_{if}$ encoding the segment-level power for each frequency bin $f$. Analogously, from Equation 1, $Z_{ij}$ represents the average count of symbolic pattern $j$ within segment $i$, derived from the columns of $\mathcal{O}$, which are orthogonal, approximately paralleling $S_{if}$ as a segment-level "power" measure. This structural analogy illustrates the similarity between the symbolic-linear composition matrix and spectrogram representations, both encoding the presence and intensity of discrete components—symbolic patterns and spectral frequencies, respectively—across temporal segments. While this architectural correspondence offers valuable intuition, it is important to note that the underlying mathematical principles of these methodologies are fundamentally distinct.

*Generality of $\boldsymbol{Z}$ with other discretizers,* $h(\cdot)$. In this case, we have chosen $h(\cdot)$ based on Lin et al. (2007). We explore other strategies such as Adaptive Brownian Bridge-based Approximation (ABBA) (Elsworth & Güttel, 2020) and Symbolic Fourier Approximation (SFA) (Schäfer & Högqvist, 2012) to construct the categorical representation and follow the same operations to obtain $\boldsymbol{Z}$. On a synthetic dataset, our explainability scores are almost similar across different discretizers (0.94 AUPRC score, approximately 10% better than the next best model; see Table 1 in Section 3.1). More datasets, evaluation metrics, and further results are given in Sections 3.1 and D.2. This highlights the generality of constructing the composition matrix $Z$ via symbolic representation $\mathcal{O}$ for better explainability. In the next section, we show how this matrix enables the construction of a global interaction-based representation that enhances temporal attribution scores.

### 2.2.3 MODULE III: GLOBAL INTERACTION OF TEMPORAL SEGMENTS

Modules I and II capture local temporal patterns through segmentation, producing latent segment embeddings $\boldsymbol{Q} \in \mathbb{R}^{\kappa \times q}$ and symbolic composition vectors $\boldsymbol{Z} \in \mathbb{R}^{\kappa \times (n \cdot v)}$, respectively. Now to predict the target label $y_i$, it is important to consider possible interactions among different segments, and those interactions can be captured by constructing a cross-representation matrix $\boldsymbol{P} = \boldsymbol{Z}^{\top}\boldsymbol{Q}$, where $\boldsymbol{P} \in \mathbb{R}^{(n \cdot v) \times q}$. $\boldsymbol{P}$ aggregates the linear relationships between symbolic and latent segment features, and its size is independent of the sequence length $L$; thereby assisting in making models with fewer trainable parameters (model details are demonstrated in Appendix B).

Each element $\mathrm{P}_{ij}$ of the matrix can be expressed as:

$$\mathrm{P}_{ij} = \sum_{k=1}^{\kappa} Z_{ki} \cdot Q_{kj}, \tag{2}$$

where $Z_{ki}$ is the $i^{\text{th}}$ symbolic feature of the $k^{\text{th}}$ segment and $Q_{kj}$ is the $j^{\text{th}}$ latent feature of the same segment. Thus, each entry in $\boldsymbol{P}$ represents a linear weighted contribution from all temporal segments, providing a global weighted summary of the time-series input. After the construction of $\boldsymbol{P}$ matrix, the temporal ordering is lost. For datasets where the discriminative features are more localized (e.g., localized frequency patterns), $\boldsymbol{Z}$ and $\boldsymbol{Q}$ are sufficient for effective classification. However, for tasks requiring explicit temporal ordering to capture sequential dependencies, we add sinusoidal positional encodings (Vaswani et al., 2017) to $\mathbf{x}_i$ in Module I (Section 2.2.1) to calculate $\boldsymbol{Q}$, while $\boldsymbol{Z}$ remains unaffected. The inclusion of positional encoding for each dataset is determined empirically by comparing predictive performance with and without it. Implementation details are provided in Appendix B.3.

**Supporting Multiple Segment Sizes.** The 2D representation $\boldsymbol{P} \in \mathbb{R}^{(n \cdot v) \times q}$, derived previously, corresponds to a fixed segment size $m$. However, a single segment size may not capture the diverse temporal patterns needed to accurately predict the output label $y_i$. To address this, we extend the computation of $\boldsymbol{P}$ to multiple segment sizes $\{m_1, m_2, \ldots, m_{|m|}\}$, and stack them along a new axis to obtain a 3D tensor: $\mathcal{R} \in \mathbb{R}^{(n \cdot v) \times q \times |m|}$, where each slice $\boldsymbol{P}^{(m_\ell)} \in \mathbb{R}^{(n \cdot v) \times q}$ is computed as described in Section 2.2.3, and $|m|$ denotes the number of distinct segment sizes. Although $\mathcal{R}$ is a 3D tensor, its dimensions do not reflect a spatial topology; hence, we do not apply any convolutional operations across this representation.

**Intuition of constructing $\boldsymbol{P}$.** Prior works such as SAX-VSM (Senin & Malinchik, 2013) demonstrate the effectiveness of representing each time-series sample as an unordered set (*Bag-of-Words*) of symbolic patterns and leveraging discriminative statistics of symbol occurrences, which enhance predictive performance. Motivated by this, our approach, in contrast, utilizes the collection of segment-wise symbolic occurrences across an input sample—i.e., the rows of $\boldsymbol{Z}$—as *a Bag-of-Stencils* (as shown in Figure 2(c)). Through the linear aggregation in Equation 2, each entry of $\boldsymbol{P}$

aggregates segments weighted by symbolic pattern occurrence (a stencil), which masks the segments where a symbol is absent and enhances the ones where it occurs more frequently, thereby modulating the corresponding latent features from $\boldsymbol{Q}$. This formulation enables $\boldsymbol{P}$ to capture relevant global discriminative interactions across the sequence linearly. This serves as the foundation for the temporal attribution scoring detailed in the following section while maintaining predictive performance.

**Training `TimeSliver`.** We obtain the predicted logits $\hat{y}_i$ by projecting the representation $\boldsymbol{P}$ to the output space using a linear layer parameterized by $\theta_c$, given as $\hat{y}_i = \mathbf{h}(\mathbf{g}(\mathbf{x}_i; \theta_q); \theta_c) \in \mathbb{R}^C$. Here, $\mathbf{g}(\cdot; \theta_q)$ represents the latent representation learning from Module I (Section 2.2.1), and $\mathbf{h}(\cdot; \theta_c)$ is the linear projection layer. We optimize the overall model by minimizing the cross-entropy loss:

$$(\theta_q^*, \theta_c^*) = \arg\min_{\theta_q, \theta_c} \sum_{i=0}^{N-1} \mathcal{L}\left(\mathrm{softmax}(\hat{y}_i), y_i\right).$$

### 2.2.4 CALCULATING TEMPORAL ATTRIBUTION

Using the optimized parameters $(\theta_q^*, \theta_c^*)$, we construct $\boldsymbol{P}$ from the symbolic composition matrix $\boldsymbol{Z}$ and learned latent representation $\boldsymbol{Q}$ (Equation 2). The matrix $\boldsymbol{P}$ captures global discriminatory features through linear operations. We then compute the positive and negative temporal attribution scores using the non-parametric function $f_{\mathrm{att}}$: $\{\phi_k^+, \phi_k^-\}_{k=1}^{L} = f_{\mathrm{att}}(\boldsymbol{P}, \boldsymbol{Z}, \boldsymbol{Q}, \hat{y})$. To explain the temporal attribution, we consider the case where $|m| = 1$, such that $\mathcal{R} \equiv \boldsymbol{P}$, although it can be extended to $|m| > 1$ without loss of generality.

Let $\hat{y}_c$ denote the logit output corresponding to the predicted class label, $c$, of the input $\mathbf{x}$. We first compute the influence of the element $P_{ij} \in \boldsymbol{P}$ on $\hat{y}_c$ as $g_{ij} = \frac{\partial \hat{y}_c}{\partial P_{ij}}$. We then define the gradient directionality of $g_{ij}$ by $\sigma_{ij} = \mathrm{sign}(g_{ij})$, which indicates whether perturbations in $P_{ij}$ are expected to increase ($\sigma_{ij} = +1$) or decrease ($\sigma_{ij} = -1$) the logit. Based on Equation 2, each $P_{ij}$ can be decomposed into $\kappa$ components corresponding to $\kappa$ temporal segments. Therefore, we estimate the normalized positive and negative contributions of the $k^{th}$ ($k \in [0, \kappa-1]$) segment for a given $g_{ij}$ and $\sigma_{ij}$ as:

$$\zeta_{k,ij}^+(g_{ij}, \sigma_{ij}) = |g_{ij}| \times \frac{\mathrm{ReLU}\left(\sigma_{ij} Z_{ki} Q_{kj}\right)}{\max_l \mathrm{ReLU}\left(\sigma_{ij} Z_{li} Q_{lj}\right) + \epsilon}, \quad \zeta_{k,ij}^-(g_{ij}, \sigma_{ij}) = |g_{ij}| \times \frac{\mathrm{ReLU}\left(-\sigma_{ij} Z_{ki} Q_{kj}\right)}{\max_l \mathrm{ReLU}\left(-\sigma_{ij} Z_{li} Q_{lj}\right) + \epsilon};$$
(3)

where $\epsilon = 10^{-18}$ and used for numerical stability. We further justify the choice of ReLU in assigning positive and negative attribution scores in Table 13 of the Appendix. While determining the contributions, it is crucial that the $Z_{ki} Q_{kj}$ terms in Equation 3 remain agnostic to the absolute scale of the terms; otherwise, this can lead to spurious attributions caused by high-magnitude but semantically irrelevant input segments. Our construction of the $\boldsymbol{Z}$ matrix, composed of the frequency of symbolic component occurrences within a segment, helps in determining a scale-invariant attribution score. A more formal representation of this property is provided in Appendix A. In Appendix F, we also show that `TimeSliver` aligns with some key desirable properties for attribution methods (Sundararajan et al., 2017). This design choice is validated through an experiment in which replacing the symbolic composition matrix $\boldsymbol{Z}$ with a dimensionality-matched projection of raw inputs $\mathbf{x}_i \in \boldsymbol{X}$ results in a 17% average drop in explainability (AUPRC) across four synthetic datasets (Figure 3a), while maintaining equivalent predictive accuracy (Figure 3b).

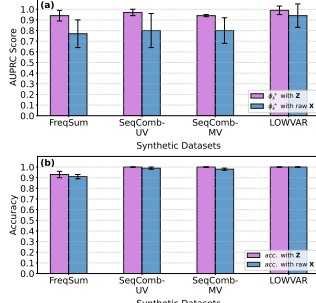

Figure 3: Impact of using raw $\boldsymbol{X}$ instead of $\boldsymbol{Z}$ on (a) explainability (AUPRC) and (b) predictability (Accuracy).

Since, $\boldsymbol{P} \in \mathbb{R}^{(n \cdot v) \times q}$ is a 2D matrix, the final positive ($\phi_k^+$) and negative ($\phi_k^-$) attribution score of the $k^{th}$ temporal segment in $x_c$ is calculated as:

$$\phi_k^+ = \sum_{i=0}^{n.v-1} \sum_{j=0}^{q-1} \zeta_{k,ij}^+ \quad \text{and} \quad \phi_k^- = \sum_{i=0}^{n.v-1} \sum_{j=0}^{q-1} \zeta_{k,ij}^-$$
(4)

### 2.2.5 METRICS TO EVALUATE TEMPORAL ATTRIBUTION

**Evaluating on synthetic dataset.** The salient time points in synthetic datasets are known (Queen et al., 2023; Liu et al., 2024) and represented by a binary vector $G \in \{0,1\}^{L \times 1}$. Temporal attribution

Table 1: Comparison of mean$_{\pm \text{std}}$ AUPRC on synthetic datasets. **Bold**: best, underlined: second-best. NA: not applicable

| Method | FreqSum | SeqComb-UV | SeqComb-MV | LOWVAR |
|---|---|---|---|---|
| Random | $0.35_{\pm 0.06}$ | $0.23_{\pm 0.04}$ | $0.22_{\pm 0.04}$ | $0.08_{\pm 0.03}$ |
| Grad-CAM | $0.64_{\pm 0.09}$ | $0.61_{\pm 0.02}$ | $0.61_{\pm 0.02}$ | $0.55_{\pm 0.01}$ |
| LIME | $0.36_{\pm 0.09}$ | $0.26_{\pm 0.07}$ | $0.23_{\pm 0.07}$ | $0.10_{\pm 0.06}$ |
| LIMESegment | NA | $0.76_{\pm 0.08}$ | NA | NA |
| KernelShap | $0.37_{\pm 0.06}$ | $0.30_{\pm 0.05}$ | $0.24_{\pm 0.05}$ | $0.12_{\pm 0.05}$ |
| Integrated Gradient | $0.59_{\pm 0.10}$ | $0.36_{\pm 0.16}$ | $0.36_{\pm 0.13}$ | $0.73_{\pm 0.34}$ |
| GradientSHAP | $0.54_{\pm 0.09}$ | $0.57_{\pm 0.09}$ | $0.39_{\pm 0.16}$ | $0.50_{\pm 0.20}$ |
| DeepLift | $0.61_{\pm 0.08}$ | $0.61_{\pm 0.03}$ | $0.57_{\pm 0.10}$ | $0.54_{\pm 0.06}$ |
| DeepLiftShap | $0.61_{\pm 0.08}$ | $0.61_{\pm 0.04}$ | $0.58_{\pm 0.09}$ | $0.54_{\pm 0.05}$ |
| Attention Tracing | $0.35_{\pm 0.06}$ | $0.24_{\pm 0.06}$ | $0.23_{\pm 0.05}$ | $0.08_{\pm 0.03}$ |
| Grad-SAM | $\underline{0.67}_{\pm 0.03}$ | $0.61_{\pm 0.02}$ | $0.61_{\pm 0.02}$ | $0.54_{\pm 0.01}$ |
| COLOR | $0.53_{\pm 0.13}$ | $\underline{0.90}_{\pm 0.05}$ | $0.72_{\pm 0.13}$ | $\underline{0.96}_{\pm 0.09}$ |
| TimeX++ | $0.59_{\pm 0.01}$ | $0.85_{\pm 0.02}$ | $\underline{0.76}_{\pm 0.01}$ | $0.95_{\pm 0.01}$ |
| TimeSliver | $\mathbf{0.94}_{\pm 0.05}$ | $\mathbf{0.97}_{\pm 0.03}$ | $\mathbf{0.94}_{\pm 0.01}$ | $\mathbf{0.99}_{\pm 0.04}$ |

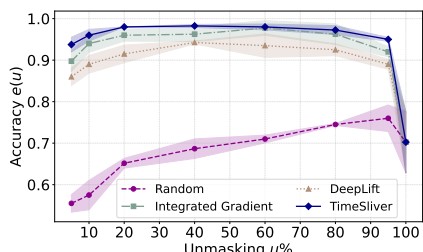

Figure 4: Positive attribution study. Accuracy curves $e(u)$ plotted against the unmasking percentage $u\%$ for EEG dataset.

scores ($\phi^+$) are softmax-normalized into probabilities and evaluated against $G$ using the area under the precision-recall curve (AUPRC), where higher values indicate a better method.

**Evaluating on real-world datasets.** To quantitatively evaluate temporal attribution scores and compare them against contemporary methods, we adapt masking-based evaluation techniques from prior work in time series (Queen et al., 2023), and protein (Pandey et al., 2025). To evaluate positive attributions, time points are first ranked based on $\phi^+$. All the time points except the top $u\%$ are then masked ($\mathbf{x}_i^t$=0 if masked) in both the training and test sets. The model is re-trained and evaluated using only the unmasked time points. Given that all datasets are class-balanced (see Appendix C for details), we use accuracy as the evaluation metric. Training quality with partial unmasking is sensitive to the masking method (zeroing or imputation) (Hooker et al., 2019). To ensure a fair comparison across explainable methods, re-training is performed on four different architectures, and the mean accuracies are reported. The value of $u$ is incrementally increased, and with each step, the model is re-trained and the accuracy $e(u)$ on the test data is recorded. The area under the $e(u)$ versus $u$ curve, $\mathcal{I}$, calculated as:

$$\mathcal{I}(\texttt{U}) = \int_0^\texttt{U} e(u)du, \tag{5}$$

is used to quantitatively compare different interpretable models. We use $\mathcal{I}(\texttt{100})$ and $\mathcal{I}(\texttt{20})$ for the comparison in our experiments. The former captures the entire area under the curve, reflecting overall explainability, while the latter emphasizes the model's effectiveness in identifying the most critical time points. The higher these values, the more interpretable the method. Unmasking time points from best to worst is more appropriate for time series data, as the discriminative information in time series is often distributed across many timesteps (Queen et al., 2023).

To assess negative attributions, we mask the top 2% and 5% of time points with the largest $\phi^-$. We expect that masking the negatively attributing time points leads to an increase in the predicted class ($\hat{y}_c$). Therefore, to compare the models in terms of their effectiveness in estimating the negatively attributing time points, we calculate: $\Delta \hat{y}_c(u^-) = \hat{y}_c(u^-) - \hat{y}_c$, where $\hat{y}_c(u^-)$ denotes the predicted logit after masking the top $u^- \in \{2\%, 5\%\}$ of negatively attributing time points. The higher the $\Delta \hat{y}_c(u^-)$ value, the better the model is at identifying the negatively attributing time points for the predicted class.

## 3   EXPERIMENTAL RESULTS

We evaluate `TimeSliver` against twelve temporal attribution methods across 7 datasets, using the explainability metrics from Section 2.2.5. We also report its accuracy on the UEA benchmark to demonstrate predictive performance.

**Datasets.** We use *four synthetic datasets* from Turbé et al. (2023) and Queen et al. (2023): FreqSum, SeqComb-UV, SeqComb-MV, and LowVar which capture a wide variety of temporal dynamics within univariate and multivariate settings (more details in Section B.1 of the Appendix). We leverage three real-world TSC applications: (1) single-channel electroencephalogram (EEG) data for sleep stage classification from 20 healthy individuals, with a sequence length of 3000; (2) the FordA machine fault diagnosis dataset for binary classification (Bagnall et al., 2018), with a sequence length of 500; and (3) an animal sound classification dataset from the Environmental Sound Classification corpus (ESC-50) (Piczak), consisting of 5-second audio clips. The audio data is processed using mel-frequency spectral representation, following standard practice (Piczak). More details about datasets are provided in Appendix B.

**Baselines.** `TimeSliver` is compared with several gradient-, kernel-, and sampling-based post-hoc methods—Grad-CAM (Selvaraju et al., 2017), DeepLIFT (Shrikumar et al., 2017), Integrated Gradients (Sundararajan et al., 2017), GradientSHAP (Lundberg & Lee, 2017), DeepLiftSHAP (Lundberg & Lee, 2017), LIME (Ribeiro et al., 2016), as well as LIMESegment (Sivill & Flach, 2022), a time-series extension of LIME, KernelSHAP (Lundberg & Lee, 2017), and TimeX++ (Liu et al., 2024), an information-bottleneck-based explainable approach for time series data. Note that LIMESegment is applicable only to univariate time series with sequence length $< 500$. We also include self-attention-based explainable models—Attention Tracing (Wu et al., 2020), which estimates temporal importance from Transformer attention weights, and Grad-SAM (Barkan et al., 2021), adapted from the language domain—as well as another explainable model from the protein domain, COLOR (Pandey et al., 2025). A `Random` baseline, assigning uniform attribution scores across time points, is also included.

**Explainability Study.** Each dataset has three distinct splits (80% train, 10% valid and 10% test), with three trials per split. For each split, we first train the predictive model using four different backbones: 1D CNN, Transformer, COLOR, and `TimeSliver` (details in Appendix B). `TimeSliver` achieves predictive performance within 3–4% of the other backbones, indicating that all models are trained comparably well and enabling a fair comparison. Subsequently, Attention Tracing and Grad-SAM are implemented on the Transformer backbone, while all other explainable methods, except COLOR, are applied to the CNN backbone. We evaluate the temporal attribution scores computed using different explainable methods on all four backbones using the metrics discussed in Section 2.2.5. The results are then averaged across all backbones for each explainable method.

## 3.1 IMPROVEMENT OF TIMESLIVER OVER BASELINES ON EXPLAINABILITY (TEMPORAL ATTRIBUTIONS)

Table 1 reports the AUPRC values computed as described in Section 2.2.5 for various explainable methods for all four synthetic datasets. **TimeSliver achieves an average of 18% improvement over the leading baseline**. Qualitative comparisons of `TimeSliver`'s temporal attribution scores against ground truth importance scores are provided in Appendix G. To quantitatively evaluate different explainable methods on real-world datasets, we report $\mathcal{I}(100)$ and $\mathcal{I}(20)$ values, as defined in Section 2.2.5, computed across three splits for all three datasets (Table 2 and Appendix D). **TimeSliver consistently outperforms baselines by 2% in $\mathcal{I}(20)$, demonstrating superior ability to identify key positive time points.** To further demonstrate the effectiveness of `TimeSliver` in capturing positive critical time steps, we present the $e(u)$ versus $u\%$ curve for the EEG dataset in Figure 4. `TimeSliver` outperforms the strongest baselines, namely Integrated Gradients and DeepLIFT.

Interestingly, Figure 4 shows a sharp accuracy drop when all time steps are unmasked ($e(100)$), revealing negatively contributing segments. Table 3 compares `TimeSliver` with baselines on computing negative temporal attributions using $\Delta \hat{y}_c(u^-)$ (Section 2.2.5). On EEG, `TimeSliver` achieves a mean increase of 0.26 in the predicted logit after masking the top negatively attributing time points, an increase that is 60% higher than the next best baseline. In the audio and FordA

Table 2: Positive attribution results, with the mean$_{\pm\text{std}}$ $\mathcal{I}(100)$ and $\mathcal{I}(20)$ values. **Bold**: best, underlined: second-best. ↑ denotes higher is better. NA: not applicable

| Method | Audio | | EEG | | FORD-A | |
|---|---|---|---|---|---|---|
| | $\mathcal{I}(\mathbf{100})$↑ | $\mathcal{I}(\mathbf{20})$↑ | $\mathcal{I}(\mathbf{100})$↑ | $\mathcal{I}(\mathbf{20})$↑ | $\mathcal{I}(\mathbf{100})$↑ | $\mathcal{I}(\mathbf{20})$↑ |
| Random | $67.90_{\pm0.30}$ | $9.06_{\pm0.02}$ | $62.66_{\pm1.85}$ | $9.33_{\pm0.38}$ | $73.89_{\pm1.96}$ | $8.89_{\pm0.25}$ |
| Grad-CAM | $69.79_{\pm0.38}$ | $10.05_{\pm0.07}$ | $67.23_{\pm0.96}$ | $10.70_{\pm0.33}$ | $81.43_{\pm0.04}$ | $11.07_{\pm0.04}$ |
| LIME | $69.09_{\pm0.44}$ | $9.54_{\pm0.37}$ | $64.02_{\pm1.3}$ | $10.55_{\pm0.33}$ | $66.92_{\pm0.46}$ | $10.47_{\pm0.04}$ |
| LIMESegment | NA | NA | NA | NA | $76.14_{\pm0.53}$ | $10.57_{\pm0.11}$ |
| KernelShap | $72.26_{\pm1.71}$ | $10.52_{\pm0.51}$ | $66.92_{\pm0.46}$ | $10.47_{\pm0.04}$ | $67.80_{\pm1.4}$ | $11.10_{\pm0.37}$ |
| Integrated Gradient | $63.69_{\pm0.04}$ | $8.34_{\pm0.46}$ | $\underline{83.19}_{\pm0.89}$ | $\underline{14.24}_{\pm0.29}$ | $\underline{93.65}_{\pm0.19}$ | $\underline{14.76}_{\pm0.14}$ |
| GradSHAP | $69.53_{\pm0.31}$ | $10.48_{\pm0.17}$ | $63.76_{\pm2.27}$ | $10.41_{\pm0.26}$ | $78.54_{\pm0.64}$ | $11.44_{\pm0.03}$ |
| DeepLift | $63.35_{\pm0.39}$ | $8.15_{\pm0.11}$ | $80.43_{\pm0.70}$ | $13.63_{\pm0.32}$ | $93.11_{\pm0.08}$ | $14.41_{\pm0.07}$ |
| DeepLiftShap | $70.55_{\pm0.40}$ | $10.70_{\pm0.08}$ | $65.34_{\pm1.78}$ | $10.66_{\pm0.34}$ | $85.30_{\pm0.27}$ | $13.30_{\pm0.07}$ |
| Attention Tracing | $69.15_{\pm0.49}$ | $9.75_{\pm0.42}$ | $63.16_{\pm2.42}$ | $9.28_{\pm0.30}$ | $76.47_{\pm0.36}$ | $9.60_{\pm0.42}$ |
| Grad-SAM | $69.00_{\pm0.12}$ | $9.89_{\pm0.24}$ | $62.11_{\pm2.92}$ | $9.38_{\pm0.34}$ | $74.17_{\pm0.01}$ | $9.17_{\pm0.14}$ |
| COLOR | $71.46_{\pm0.67}$ | $10.47_{\pm0.14}$ | $66.48_{\pm1.47}$ | $10.85_{\pm0.34}$ | $83.95_{\pm0.85}$ | $12.12_{\pm0.22}$ |
| TimeX++ | $\underline{73.24}_{\pm0.35}$ | $\underline{11.20}_{\pm0.12}$ | $74.10_{\pm0.49}$ | $11.84_{\pm0.09}$ | $87.85_{\pm0.53}$ | $13.83_{\pm0.18}$ |
| TimeSliver | $\mathbf{74.30}_{\pm0.68}$ | $\mathbf{11.35}_{\pm0.15}$ | $\mathbf{83.99}_{\pm0.61}$ | $\mathbf{14.52}_{\pm0.15}$ | $\mathbf{93.87}_{\pm0.01}$ | $\mathbf{14.99}_{\pm0.01}$ |

Table 3: Negative attribution results showing $\Delta\hat{y}_c(u^-)$ . **Bold**: best; underlined: second-best. ↑ indicates higher is better. NA: not applicable

| Methods | EEG | | Audio | | FordA | |
|---|---|---|---|---|---|---|
| | $\Delta\hat{y}_c(2\%)\uparrow$ | $\Delta\hat{y}_c(5\%)\uparrow$ | $\Delta\hat{y}_c(2\%)\uparrow$ | $\Delta\hat{y}_c(5\%)\uparrow$ | $\Delta\hat{y}_c(2\%)\uparrow$ | $\Delta\hat{y}_c(5\%)\uparrow$ |
| Random | $-0.11_{\pm0.2}$ | $-0.15_{\pm0.19}$ | $-0.25_{\pm0.27}$ | $-0.20_{\pm0.22}$ | $-0.04_{\pm0.17}$ | $-0.08_{\pm0.25}$ |
| Grad-CAM | $0.04_{\pm0.11}$ | $0.01_{\pm0.12}$ | $-0.07_{\pm0.23}$ | $-0.12_{\pm0.28}$ | $-0.03_{\pm0.18}$ | $-0.09_{\pm0.27}$ |
| LIME | $0.05_{\pm0.10}$ | $0.06_{\pm0.11}$ | $-0.10_{\pm0.20}$ | $-0.10_{\pm0.19}$ | $-0.08_{\pm0.21}$ | $-0.10_{\pm0.27}$ |
| LIMESegment | NA | NA | NA | NA | $\mathbf{0.02}_{\pm0.07}$ | $\mathbf{0.02}_{\pm0.12}$ |
| KernelShap | $0.06_{\pm0.11}$ | $0.03_{\pm0.12}$ | $-0.17_{\pm0.35}$ | $-0.16_{\pm0.31}$ | $-0.04_{\pm0.18}$ | $-0.13_{\pm0.23}$ |
| Integrated Gradient | $\underline{0.16}_{\pm0.17}$ | $\underline{0.17}_{\pm0.20}$ | $-0.12_{\pm0.27}$ | $\underline{-0.08}_{\pm0.28}$ | $\mathbf{0.02}_{\pm0.08}$ | $\mathbf{0.02}_{\pm0.07}$ |
| GradSHAP | $0.00_{\pm0.17}$ | $-0.02_{\pm0.15}$ | $-0.22_{\pm0.30}$ | $-0.18_{\pm0.31}$ | $-0.05_{\pm0.17}$ | $-0.04_{\pm0.15}$ |
| DeepLift | $0.11_{\pm0.17}$ | $0.16_{\pm0.17}$ | $-0.15_{\pm0.33}$ | $-0.17_{\pm0.30}$ | $0.00_{\pm0.16}$ | $\mathbf{0.02}_{\pm0.07}$ |
| DeepLiftShap | $0.03_{\pm0.16}$ | $0.07_{\pm0.17}$ | $-0.15_{\pm0.31}$ | $-0.14_{\pm0.30}$ | $0.00_{\pm0.10}$ | $0.00_{\pm0.14}$ |
| Attention Tracing | $0.06_{\pm0.13}$ | $-0.06_{\pm0.13}$ | $-0.13_{\pm0.27}$ | $-0.13_{\pm0.29}$ | $-0.05_{\pm0.19}$ | $-0.11_{\pm0.17}$ |
| Grad-SAM | $0.06_{\pm0.13}$ | $0.00_{\pm0.16}$ | $-0.16_{\pm0.31}$ | $-0.11_{\pm0.26}$ | $-0.10_{\pm0.26}$ | $-0.05_{\pm0.21}$ |
| COLOR | $0.05_{\pm0.14}$ | $0.03_{\pm0.14}$ | $-0.19_{\pm0.25}$ | $\underline{-0.08}_{\pm0.22}$ | $-0.04_{\pm0.16}$ | $-0.08_{\pm0.21}$ |
| TimeX++ | $0.12_{\pm0.16}$ | $0.11_{\pm0.12}$ | $\underline{-0.09}_{\pm0.23}$ | $-0.15_{\pm0.33}$ | $-0.01_{\pm0.17}$ | $-0.05_{\pm0.15}$ |
| TimeSliver | $\mathbf{0.26}_{\pm0.22}$ | $\mathbf{0.26}_{\pm0.19}$ | $\mathbf{-0.07}_{\pm0.22}$ | $\mathbf{-0.08}_{\pm0.26}$ | $\mathbf{0.02}_{\pm0.07}$ | $\mathbf{0.02}_{\pm0.07}$ |

datasets, the mean increase in the logit is close to zero, indicating the absence of negatively attributing time points in the majority of samples.

## 3.2 COMPETITIVE PERFORMANCE OF TimeSliver ON MULTIVARIATE TIME SERIES CLASSIFICATION

We evaluate the predictive performance of TimeSliver ($f_{cls}$) on 26 datasets from the UEA multivariate time-series classification archive (Ruiz et al., 2021), which span 8 electrical biosignal (Bio.) datasets, 3 audio datasets, 7 accelerometer-based motion datasets, 3 gesture and digit recognition datasets in Cartesian coordinates (Coord.), and other miscellaneous datasets and evaluate TimeSliver against five methodological categories: *Distance-based* methods (Bagnall et al., 2016); *Dictionary/interval-based* methods (Schäfer & Leser, 2017); *Feature-based ML* models; *Deep learning* models including ResNet (Wang et al., 2017), InceptionTime (Fawaz et al., 2020), FCN (Karim et al., 2017), TS2vec (Yue et al., 2022), TimesNet (Wu et al., 2022), ShapeNet (Li et al., 2021b), RL-PAM (Gao et al., 2022b), ShapeConv (Qu et al., 2024b), SBM (Wen et al., 2025a), InterpGN (Wen et al., 2025a); and *Ensemble-based* methods. More details are provided in Section E of

Table 4: Accuracy (acc.) of TimeSliver vs. 16 baselines on 26 UEA datasets. **Bold**: best, Underlined: second-best.

| Type | Method | Bio. | Motion | Audio | Coord. | Misc. | All |
|---|---|---|---|---|---|---|---|
| Distance-Based | DTW_D | 46.9 | 87.5 | 44.5 | 95.4 | 63.9 | 66.8 |
| | DTW_I | 47.9 | 77.1 | 34.4 | 90.7 | 59.2 | 61.3 |
| | DTW_A | 45.4 | 88.1 | **63.3** | 95.4 | 71.3 | 69.7 |
| Dictionary-Based | MUSE | 55.7 | 88.3 | 51.8 | 96.0 | 81.7 | 72.3 |
| | gRSF | 49.0 | 84.3 | 46.1 | 88.3 | 71.7 | 63.8 |
| | CIF | 54.0 | 85.4 | 55.1 | 96.2 | **83.0** | 71.5 |
| Feature-Based ML | MrSEQL | 52.3 | 87.8 | 47.6 | 94.2 | 76.6 | 69.2 |
| | ROCKET | 53.8 | 90.1 | 48.7 | 96.6 | 75.6 | 70.2 |
| Deep-Learning | TapNet | 49.9 | 84.5 | 51.5 | 91.5 | 65.4 | 64.8 |
| | ResNet | 48.3 | 90.6 | 47.5 | 97.3 | 57.2 | 63.3 |
| | IncTime | 60.3 | **96.4** | 61.6 | 95.0 | 69.1 | 74.3 |
| | FCN | 56.76 | 90.8 | 58.4 | 97.8 | 63.4 | 72.2 |
| | TS2Vec | 48.57 | 87.4 | 53.2 | 94.7 | 65.0 | 68.0 |
| | TimesNet | 61.06 | 77.9 | 42.1 | 91.3 | 61.9 | 67.3 |
| | ShapeNet | 55.28 | 86.3 | 59.3 | 94.0 | 53.3 | 68.2 |
| | RLPAM | 66.75 | 89.5 | 55.1 | 90.0 | 67.4 | 74.3 |
| | ShapeConv | 61.24 | 89.0 | 54.1 | 95.0 | 64.8 | 72.4 |
| | SBM | 59.8 | 86.0 | 45.8 | 94.1 | 66.7 | 70.5 |
| | InterpGN | 58.73 | 91.4 | 52.6 | **98.3** | 70.7 | 73.7 |
| **Ours** | TimeSliver | **66.9** | 90.9 | 55.2 | 93.6 | 76.1 | **75.6** |

the Appendix. We report the predictive performance in Table 4 and observe that: (1) TimeSliver demonstrates **superior performance on datasets with long sequences** (length > 1000), achieving **an average absolute improvement of 4.4%** over the best-performing baselines (details in Appendix E); (2) it **improves performance on electrical biosignals by 6.3% on average**; and (3) overall, TimeSliver delivers **competitive predictive performance, staying within 2% of the best baselines** across diverse applications. To further showcase TimeSliver's strong performance, we report its average rank, top-1, and top-3 counts compared to all baselines in Appendix E.

To empirically demonstrate TimeSliver's ability to capture temporally disjoint yet jointly informative patterns, we construct a synthetic dataset with multiplicative interactions in disjoint segments (Appendix D.3) and evaluate its predictive performance against multiple architectures. As shown

in Appendix D.3, TimeSliver achieves performance within 1% of the baselines, confirming its effectiveness in modeling temporally disjoint interactions.

### 3.3 UNDERSTANDING THE COMPONENTS OF TIMESLIVER

We assess the impact of TimeSliver's core design choices through ablation and sensitivity analysis to gain deeper insights.

**Ablation Study** (Figure 5). The framework of Boureau et al. (2010) shows that average pooling lowers feature map resolution, reducing sensitivity to local perturbations and acting as a regularizer. We verify this by removing average pooling after $\mathcal{R}$ or replacing it with max pooling, which causes a $\sim$5% drop in accuracy with negligible impact on explainability. To assess the role of ReLU in Equations 4 in identifying positive and negative attributing time points, we replace ReLU with abs. This assigns equal importance to time points with equal $\mathrm{abs}(Z_{ki}Q_{kj})$ but opposite signs. This modification leads to a 13% drop in explainability, underscoring the importance of ReLU in correctly distinguishing positive and negative contributions.

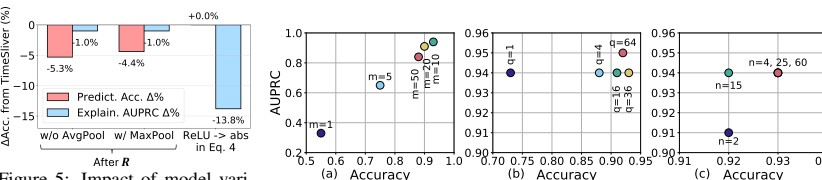

Figure 5: Impact of model variants on prediction (Predict.) and explainability (Explain.).

Figure 6: Effect of (a) segment size $m$, (b) latent dimension $q$, and (c) number of bins $n$ on predictability (Accuracy) and explainability (AUPRC); (d) GFLOPs variation with $n$ and $|m|$.

**Sensitivity Analysis** (Figure 6). The final architecture of TimeSliver is determined by three key hyperparameters: the segment size $m$, the latent representation dimension $q$ (both defined in Section 2.2.1), and the number of bins $n$ used to discretize raw inputs $\mathbf{x}_i$ into symbolic representations $s_i$ (described in Section 2.2.2). On the FreqSum dataset, Figure 6a shows that explainability declines for $m > 10$, as larger segments can lead the model to over-attribute importance to regions where only a small part is relevant. Conversely, setting $m = 1$ results in poor predictability and explainability, as the segment is too short to capture meaningful temporal patterns. The effect of the latent dimension $q$ and the bin count $n$ is minimal beyond values of 4, with both explainability and predictive accuracy remaining stable (see Figures 6b and 6c). Although this analysis is based on FreqSum data, the relative sensitivity trends are expected to generalize across a wide range of real-world datasets. Figure 6d shows that TimeSliver's GFLOPs only scale linearly with $n \in [2, 500]$ and $|m| \in [1, 5]$, remaining 5–10 times lower than those of Transformers (GFLOP = 0.2), highlighting its efficient scalability.

## 4 CONCLUSION

In this work, we presented TimeSliver—a novel deep learning framework that linearly combines raw time series with their symbolic counterparts to construct a global representation facilitating temporal attribution calculation. Our importance scores offer insights into positively and negatively influencing time segments. The effectiveness of TimeSliver is demonstrated by its average improvement of 11% over the best baselines across seven diverse datasets, spanning real-world and synthetic, univariate and multivariate time series with varied temporal dynamics, while maintaining high predictive performance. In the future, it will be interesting to consider human-in-the-loop expert validation (for tasks like sleep-stage classification using EEG) to harness TimeSliver's explainability for practical applications. Additionally, TimeSliver's principles can be extended to provide feature attribution, identifying which input features are most influential at each time segment, especially by considering a time-frequency representation of time-series data.

## 5 ACKNOWLEDGMENT

We gratefully acknowledge the support of the National Science Foundation's MRSEC program (DMR-2308691) at the Materials Research Center of Northwestern University, as well as National Science Foundation grants 2324936 and 2328973.

ETHICS STATEMENT.

This work does not involve human subjects, sensitive data, or issues related to fairness, discrimination, or legal compliance. `TimeSliver` is designed to identify influential temporal segments in time series, providing more transparent and interpretable model predictions. By improving explainability, particularly for applications such as healthcare time-series classification, `TimeSliver` supports responsible and trustworthy deployment of machine learning models.

REPRODUCIBILITY STATEMENT

All source code to reproduce experimental results (with instructions for running the code) is provided in the Supplementary Materials. We use public datasets and include implementation details in the Appendix. All baselines either adopt published hyperparameters or are tuned when unspecified.

LLM USAGE STATEMENT

The usage of LLMs in this work is limited to paper writing support, language refinement, and experimental data processing. Specifically, LLMs assisted in improving the clarity and coherence of the manuscript, generating LaTeX tables, and formatting results for presentation. Importantly, LLMs were not involved in the design of algorithms, the development of theoretical results, or the execution of experiments, ensuring that all core scientific contributions remain entirely the work of the authors.

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

# APPENDIX

This appendix provides additional details for "`TimeSliver`: Symbolic-Linear Decomposition for Explainable Time Series Classification". Additional implementation details for `TimeSliver` and the backbone models are presented in Section B. Class distribution for the four datasets used in the interpretability study are provided in Section C. Detailed results on interpretability and predictive performance are given in Sections D and E, respectively.

## A  SCALE-INVARIANCE PROPERTIES OF TIMESLIVER

**Remark (Determining Scale-Invariant Attributions).** Let $\mathbf{P}_{\text{raw}} = \mathbf{X}\mathbf{Q}$ and $\mathbf{P}_{\text{sym}} = \mathbf{Z}\mathbf{Q}$, where $\mathbf{Q}$ are learned weights. Suppose $\mathbf{X} = \mathbf{Z}\mathbf{D}$ for a diagonal scaling matrix $\mathbf{D}$. Then, for any position $t$,

$$\|\mathbf{P}_{\text{raw}}^{(t)}\|_2 = d_t \|\mathbf{P}_{\text{sym}}^{(t)}\|_2,$$

where $d_t$ is the $t$-th diagonal entry of $\mathbf{D}$. Thus, only $\mathbf{P}_{\text{sym}}$ yields attributions invariant to input scaling, and explanations depend solely on the symbolic pattern, not on the magnitude of the input.

*Implication.* This property prevents spurious attributions caused by high-magnitude but semantically irrelevant input segments, and is essential for robust and interpretable explanations. The effectiveness of this property is further validated by our ablation study, where replacing the one-hot encoding representation $\mathcal{O}$ with raw data $x$ in Equation 1 results in an average 17% decrease in explainability as shown in Figure 3.

## B  ADDITIONAL IMPLEMENTATION DETAILS

### B.1  DATASET DESCRIPTION

**FreqSum** is a multivariate time series with randomly embedded sine-wave segments; classes indicate whether the sum of their frequencies exceeds a threshold. As described in (Turbé et al., 2023), each sample in the dataset consists of 6 features and 500 time steps. To simulate realistic temporal dependencies, each feature includes a baseline sine wave with a frequency uniformly sampled from the range $[2, 5]$. Two randomly selected features per sample are injected with discriminative sine waves, each supported over 100 time steps, with frequencies drawn from a discrete uniform distribution in the range $[10, 50]$. In the remaining four features, a square wave is optionally added with 50% probability, also using frequencies sampled from the same range. The classification task is binary: the model must predict whether the sum of the two discriminative frequencies exceeds a predefined threshold, set to $\tau = 60$.

**SeqComb-UV, SeqComb-MV**, and **LowVar** are generated using the exact technique discussed in Queen et al. (2023). **SeqComb-UV** is a univariate series with two non-overlapping increasing or decreasing subsequences, with four classes defined by their trend combinations. **SeqComb-MV** is the multivariate extension of SeqComb-UV. **LowVar** is a multivariate series with four classes determined by the presence of a low-variance subsequence in a specific channel.

**Audio Dataset.** We use a manually curated subset of the ESC-50 audio dataset, focusing exclusively on animal sounds. This subset was selected to leverage the temporal localization of animal sounds, which typically occur within short bursts in the observation window, as opposed to environmental sounds that span the entire duration and yield robust results even with randomly sampled segments. This temporal sparsity makes animal sounds particularly useful for evaluating interpretability methods that rely on temporal attribution. For preprocessing, we extract Mel-frequency cepstral coefficients (MFCCs) from the audio using a Mel spectrogram with 40 Mel bands, employing standard settings such as centered windowing and normalization similar to previous works (Mohapatra et al., 2022; 2023).

**EEG Dataset.** This dataset comprises single-channel EEG recordings collected from 20 subjects, with the objective of classifying five sleep stages: wake, N1, N2, N3 (non-REM stages), and REM (rapid eye movement). The temporal structure of EEG signals makes this dataset well-suited for tasks requiring time-series modeling and interpretation. We balance all the classes in the dataset before using it for the study. We follow similar preprocessing as previous work (Mohapatra et al., 2025b).

**FordA Dataset.** We adopt the data preprocessing and train-test splits for the FordA dataset as defined in the MTS-Bakeoff benchmark Ruiz et al. (2021).

## B.2    DATASET DETAILS

Information such as the number of variates ($v$), maximum sequence length, and dataset splits is provided in Table 5.

Table 5: Summary of the four datasets used in the interpretability study.

| Dataset | Num. of Variates, $v$ | Max Seq. Length | Train | Valid | Test |
|---|---|---|---|---|---|
| FreqSum | 6 | 500 | 5000 | 500 | 500 |
| SeqComb-UV | 1 | 200 | 5000 | 1000 | 1000 |
| SeqComb-MV | 4 | 200 | 5000 | 1000 | 1000 |
| LowVar | 2 | 200 | 5000 | 1000 | 1000 |
| Audio | 40 | 501 | 280 | 60 | 60 |
| EEG | 1 | 3000 | 5005 | 1295 | 3515 |
| Ford-A | 1 | 500 | 853 | 106 | 119 |

## B.3    MODEL DETAILS

The complete details of `TimeSliver` for all four datasets are given in Table 6. The selective use of positional encoding shown in the table is determined empirically based on the predictive performance of `TimeSliver` with and without it. It is also worth noting that positional encoding is only added in Module I (Section 2.2.1), thereby only changing $Q$ and not affecting $Z$ in Equation 1. Additionally, the details for the other three backbones used in the interpretability study are given in Table 7.

Table 6: Architecture details of `TimeSliver` used for different datasets.

| Dataset | Num. of categorical bins, $n$ | Num. of columns in $\mathcal{O}$, $n \times v$ | Latent vector size, $q$ | Segment size, $m$ | Positional Encoding | Trainable parameters |
|---|---|---|---|---|---|---|
| FreqSum | 15 | 90 | 36 | 7 | ✗ | 5,858 |
| SeqComb-UV | 20 | 20 | 36 | [4,7] | ✓ | 14,518 |
| SeqComb-MV | 10 | 40 | 36 | [4,7] | ✓ | 20,576 |
| LowVar | 20 | 40 | 36 | 4 | ✗ | 5,078 |
| Audio | 10 | 400 | 12 | 1 | ✗ | 20,110 |
| EEG | 25 | 25 | 12 | 10 | ✓ | 6,441 |
| Ford-A | 70 | 70 | 36 | 10 | ✗ | 8,280 |

Table 7: Number of trainable parameters for different model architectures across datasets.

| Dataset | CNN | COLOR | Transformer |
|---|---|---|---|
| FreqSum | 42,378 | 2,660 | 46,714 |
| SeqComb-UV | 42,076 | 16,844 | 361,156 |
| SeqComb-MV | 42,268 | 21,452 | 361,540 |
| LowVar | 42,140 | 16,880 | 361,284 |
| Audio | 224,938 | 8,206 | 370,498 |
| EEG | 74,981 | 43,309 | 230,805 |
| Ford-A | 42,058 | 26,536 | 361,090 |

## B.4    TRAINING AND OPTIMIZATION DETAILS

All experiments are conducted on a server running Ubuntu OS, equipped with NVIDIA RTX A6000 GPUs, using the PyTorch framework. During model training, we employ the Adam optimizer with a

learning rate ranging from $3 \times 10^{-4}$ to $1 \times 10^{-3}$. Validation accuracy is used for early stopping and to save the best model checkpoint.

## B.5 PREDICTIVE RESULTS ON DIFFERENT BACKBONE

Table 4 presents the predictive performance of the four deep learning models used as backbones in the interpretability study. The CNN backbone is used for all post-hoc interpretability methods, while the Transformer is employed for attention tracing and the Grad-SAM method. COLOR, originally developed for protein sequence design, is inherently interpretable. The predictive performance of `TimeSliver` on the four datasets used in the interpretability study is within 3–4% of the best-performing model. All the post-hoc methods are implemented using the Captum library [0] in PyTorch.

Table 8: Accuracy (mean$_{\pm\text{std}}$) over 3 runs for different predictive backbone and dataset (supporting results for Section 3 in the main paper).

| Dataset | CNN | COLOR | Transformer | TimeSliver |
|---|---|---|---|---|
| FreqSum | $0.93_{\pm0.028}$ | $0.93_{\pm0.014}$ | $0.95_{\pm0.0071}$ | $0.93_{\pm0.014}$ |
| SeqComb-UV | $1.00_{\pm0.00}$ | $1.00_{\pm0.00}$ | $1.00_{\pm0.00}$ | $1.00_{\pm0.00}$ |
| SeqComb-MV | $1.00_{\pm0.00}$ | $1.00_{\pm0.00}$ | $1.00_{\pm0.00}$ | $1.00_{\pm0.00}$ |
| LowVar | $1.00_{\pm0.00}$ | $1.00_{\pm0.00}$ | $1.00_{\pm0.00}$ | $1.00_{\pm0.00}$ |
| Audio | $0.78_{\pm0.0071}$ | $0.80_{\pm0.021}$ | $0.80_{\pm0.000}$ | $0.81_{\pm0.0071}$ |
| EEG | $0.78_{\pm0.019}$ | $0.73_{\pm0.039}$ | $0.68_{\pm0.038}$ | $0.72_{\pm0.042}$ |
| FordA | $0.92_{\pm0.0071}$ | $0.93_{\pm0.000}$ | $0.83_{\pm0.0071}$ | $0.88_{\pm0.000}$ |

## C CLASS DISTRIBUTION

The class distribution for all four datasets is shown in Figure 7, indicating that there is no class imbalance in any of the datasets used in the explainability study.

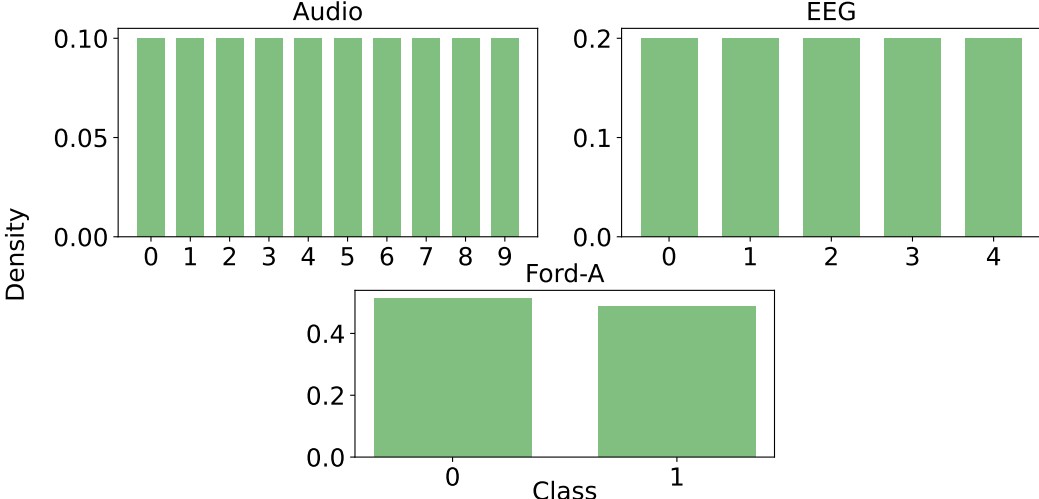

Figure 7: Class distribution among different datasets.

[0]https://github.com/pytorch/captum

## D   DETAILED EXPLAINABILITY RESULTS

### D.1   EVALUATING POSITIVE TEMPORAL ATTRIBUTION

Figure 8 shows the mean $e(u)$ versus unmasking percentage ($u\%$) curves obtained using different interpretability methods, along with their standard deviations. The trend of the curves clearly demonstrates that TimeSliver outperforms the baseline methods in the lower unmasking range (5–20%), highlighting its effectiveness in identifying the most critical time points.

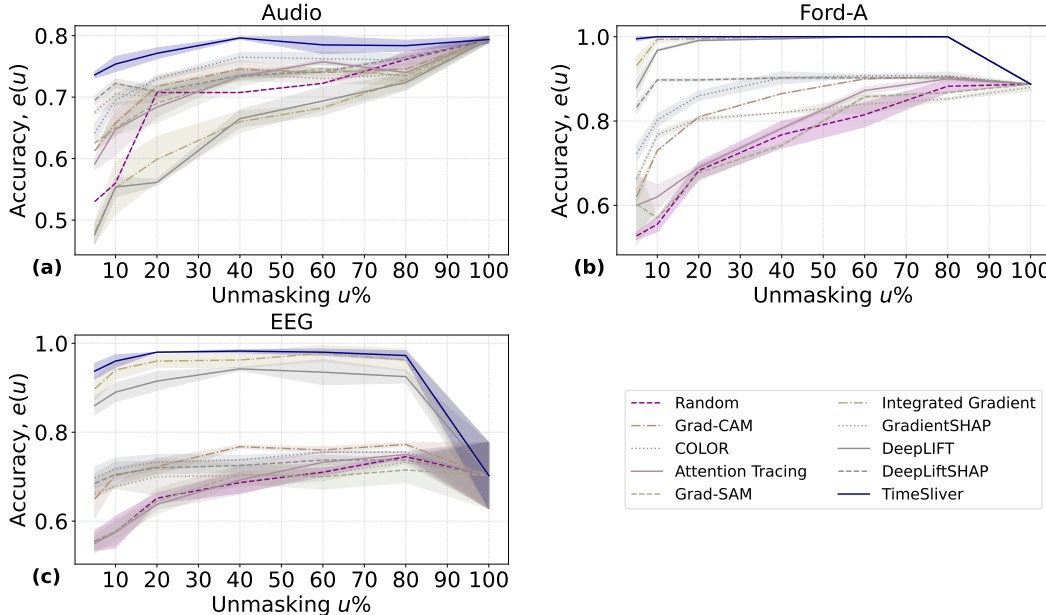

Figure 8: Positive attribution study. Accuracy curves $e(u)$ plotted against the unmasking percentage $u\%$ for various methods on three datasets: a) Audio, b) Ford-A, and c) EEG SSC. Each curve represents the mean accuracy over three runs (supporting results for Section 3.1 in the main paper).

The areas under the curves shown in Figure 8, $\mathcal{I}(100)$ and $\mathcal{I}(20)$, are used to quantitatively compare the different interpretability methods. To calculate $\Delta\hat{y}_c(u^-)$ (Section 2.2.5) and compare different models in estimating negatively attributing time points, we conduct the study across three different data splits and average the results over all samples.

### D.2   IMPACT OF OTHER SYMBOLIC REPRESENTATIONS

Table 9 demonstrates the impact of different discretization functions $h(\cdot)$ (defined in Section 2.2.1) for converting continuous time series into symbolic representations $s_i = h(\mathbf{x}_i; n, w)$. Both ABBA and SFA preserve explainability (AUPRC) and predictability (Accuracy) compared to our default binning approach. However, when we replace the symbolic representation with a higher-dimensional projection of the raw input ($\mathcal{O} \rightarrow x_{proj}$) to calculate $Z$, explainability drops significantly by 38%, while predictability remains unchanged. This demonstrates that discretizing the continuous time series $x$ into symbolic representation $s$ provides a scale-invariant encoding that ensures uniform treatment of temporal patterns regardless of input magnitude.

### D.3   TIMESLIVER'S EFFECTIVENESS IN CAPTURING FAR-FIELD INTERACTION

Although $P$ in Equation 2 is formulated as a linear aggregation of temporal segments, it does not significantly affect the ability of TimeSliver to capture far-field multiplicative interactions. To demonstrate this, we construct a synthetic dataset designed specifically to exhibit strong far-field dependencies.

| Symbolic Representation | **Explainability** AUPRC($\Delta\%$) | **Predictability** Accuracy($\Delta\%$) |
|---|---|---|
| `TimeSliver` with binning | $0.94_{\pm 0.045}$ | $0.93_{\pm 0.014}$ |
| ABBA | $0.94_{\pm 0.048}$ (0%) | $0.93_{\pm 0.05}$ (0%) |
| SFA | $0.93_{\pm 0.068}$ (-1.0%) | $0.92_{\pm 0.02}$ (-1.0%) |
| $\mathcal{O} \to x_{proj}$ | $0.66_{\pm 0.13}$ (-38.7%) | $0.91_{\pm 0.015}$ (-2.2%) |

Table 9: Impact of symbolic representations on explainability and predictability.

**Input Construction.** We generate $N = 1000$ samples, each of length $L = 100$. Each sample $\mathbf{x}_i$ is defined as:

$$\mathbf{x}_i(t) = \sin(f_i t + \phi_i) + \eta_i(t) \cdot \left(t - \frac{L}{2}\right), \quad t = 1, \ldots, L, \tag{6}$$

where:

- $f_i \sim \mathcal{U}(1, 10)$ is a randomly sampled frequency,
- $\phi_i \sim \mathcal{U}(0, 2\pi)$ is a random phase shift, and
- $\eta_i(t)$ is Gaussian noise scaled by the time component $\left(t - \frac{L}{2}\right)$ to amplify far-field interactions.

**Output Property (Far-Field Interaction).** For each sample $\mathbf{x}_i$, we define the far-field interaction property:

$$p_i = \sum_{j=1}^{L/2} \mathbf{x}_i[j] \cdot \mathbf{x}_i[L - j + 1], \tag{7}$$

where $\mathbf{x}_i[j]$ is the $j^{\text{th}}$ element and $\mathbf{x}_i[L - j + 1]$ is its *far-field pair* from the opposite end of the sequence.

A binary label $y_i$ is then assigned:

$$y_i = \begin{cases} 1, & \text{if } p_i > 0, \\ 0, & \text{if } p_i \leq 0, \end{cases} \tag{8}$$

resulting in a balanced class distribution with a 0:1 ratio of 1:1.

**Results.** Table 10 compares the predictive performance of `TimeSliver` with Transformer, LSTM, and FCN baselines. The results empirically confirm that `TimeSliver` effectively captures multiplicative far-field interactions, which we attribute to the fully connected neural network layer present after $P$ in the architecture.

| **Metric** | Transformer | LSTM | `TimeSliver` | FCN |
|---|---|---|---|---|
| **Balanced Accuracy** | 0.68 (0.014) | 0.73 (0.04) | 0.76 (0.02) | 0.77 (0.05) |

Table 10: Predictive performance on the synthetic far-field interaction dataset. Results are reported as mean (std) accuracy over three runs.

# E  DETAILED PREDICTABILITY RESULTS

We evaluate `TimeSliver` against five methodological categories: (1) *Distance-based* methods, including DTW variants (Bagnall et al., 2016); (2) *Dictionary/interval-based* methods, such as MUSE (Schäfer & Leser, 2017) and gRFS/CIF (Middlehurst et al., 2020); (3) *Feature-based ML* models like ROCKET (Dempster et al., 2020) and MrSEQL (Karlsson et al., 2016); (4) *Deep learning* models including ResNet (Wang et al., 2017), InceptionTime (Fawaz et al., 2020), FCN (Karim et al., 2017), TS2vec (Yue et al., 2022), TimesNet (Wu et al., 2022), ShapeNet (Li et al., 2021b), RLPAM (Gao et al., 2022b), ShapeConv (Qu et al., 2024b), SBM (Wen et al., 2025a), InterpGN (Wen et al., 2025a); and (5) *Ensemble-based* approaches such as CBOSS, STC (Bagnall et al., 2016; 2017), RISE,

TSF, and HC (Lines et al., 2018). The predictive performance of `TimeSliver` on all 26 UEA datasets, along with the results of the baseline methods, is presented in Table 11. The comparison of `TimeSliver` with baseline methods in terms of average rank, top-1, and top-3 counts is presented in Table 12.

## F    THEORETICALLY DESIRABLE PROPERTIES

### F.1    COMPLETENESS

To satisfy the *completeness* axiom, the attribution scores of all temporal segments for class $c$ in input $\mathbf{x}$ must sum to the change in the predicted logit for class $c$ between $\mathbf{x}$ and a baseline $\mathbf{x}_{\text{baseline}}$:

$$\hat{y}_c(\mathbf{x}) - \hat{y}_c(\mathbf{x}_{\text{baseline}}) = \sum_{k=0}^{\kappa-1} \phi_k,$$

where $\kappa$ is the number of temporal segments and $\phi$ is the attribution score. In this study, we set $\mathbf{x}_{\text{baseline}} = 0$, so that $\hat{y}_c(\mathbf{x}_{\text{baseline}}) = 0$. Hence, we need to show that

$$\hat{y}_c(\mathbf{x}) = \sum_{k=0}^{\kappa-1} \phi_k.$$

As noted in Section 2.2, `TimeSliver` transforms $\boldsymbol{P}$ into class logits using just a linear layer ($\theta_c$). Thus the predicted logit of class $c$ for input $\mathbf{x}$ can be expressed as:

$$\hat{y}_c = \sum_{i=0}^{n.\nu-1} \sum_{j=0}^{q-1} w_{ij} P_{ij} \tag{9}$$

This implies that

$$\frac{\partial \hat{y}_c}{\partial P_{ij}} = w_{ij} \tag{10}$$

Let's assume that $\mathbf{x}$ only consists of positively attributing temporal segments for class $c$. This implies that

- $\sigma_{ij} = \text{sign}(w_{ij}) = +1$ for $\forall\,(i,j) \in \{0,\dots,n.\nu-1\} \times \{0,\dots,q\}$.
- $Z_{ki}Q_{kj} >= 0\ \forall (i,j,k) \in \{0,\dots,n.\nu-1\} \times \{0,\dots,q\} \times \{0,\dots,\kappa-1\}$ in Equation 3.

Additionally, based on the empirical results presented in Table 13, we can remove max-scaling in Equation 3 by accepting a decrease in AUPRC score by 1.5%. Based on the above conditions, we can rewrite Equation 3 as:

$$\zeta_{k,ij}^+(w_{ij}) = w_{ij} \times Z_{ki}Q_{kj} \quad \text{and} \quad \zeta_{k,ij}^-(w_{ij}) = 0 \tag{11}$$

Further, the total attribution score of a $k^{th}$ temporal segment can be calculated as:

$$\phi_k^+ = \sum_{i=0}^{n.\nu-1} \sum_{j=0}^{q-1} w_{ij} \times Z_{ki}Q_{kj} \tag{12}$$

Adding the attribution scores of all the temporal segments and using Equation 2 leads to

$$\sum_{k=0}^{\kappa-1} \phi_k^+ = \sum_{i=0}^{n.\nu-1} \sum_{j=0}^{q-1} w_{ij} \times \sum_{k=0}^{\kappa-1} Z_{ki}Q_{kj} = \sum_{i=0}^{n.\nu-1} \sum_{j=0}^{q-1} w_{ij} P_{ij} = \hat{y}_c \tag{13}$$

Equation 13 shows that under the conditions discussed above, `TimeSliver` satisfies the *completeness* axiom. It is worth noting that this axiom will also be satisfied if there are only negatively attributing segments for class $c$ in $\mathbf{x}$.

| Problem | DTW_D | DTW_I | DTW_A | MUSE | gRSF | CIF | MrSEQL | ROCKET | CBOSS | STC | RISE | TSF | HC | TapNet | ResNet | InceptionTime | TimeSliver |
|---|---|---|---|---|---|---|---|---|---|---|---|---|---|---|---|---|---|
| ArticularyWordRecognition | 98.87 | 94.31 | 98.94 | 98.87 | 98.21 | 97.89 | 98.98 | **99.56** | 97.56 | 97.51 | 95.73 | 94.82 | 97.99 | 97.13 | 98.26 | 99.10 | 99.33 |
| AtrialFibrillation | 23.56 | 34.67 | 22.44 | **74.00** | 27.56 | 25.11 | 36.89 | 24.89 | 30.44 | 31.78 | 24.44 | 29.78 | 29.33 | 30.22 | 36.22 | 22.00 | 73.00 |
| BasicMotions | 95.25 | 97.17 | 99.92 | 100.00 | 99.83 | 99.75 | 94.83 | 99.00 | 98.75 | 97.92 | 100.00 | 98.78 | 100.00 | 99.08 | 100.00 | 100.00 | 100.00 |
| Cricket | 100.00 | 95.74 | 100.00 | 99.77 | 97.41 | 98.38 | 99.21 | 100.00 | 97.55 | 98.94 | 97.78 | 93.15 | 99.26 | 97.50 | 99.40 | 99.44 | 98.61 |
| DuckDuckGeese | 49.20 | 29.27 | 56.67 | 56.00 | 44.47 | 56.00 | 39.27 | 46.13 | 43.07 | 43.47 | 50.80 | 38.87 | 47.60 | 58.27 | 63.20 | 63.47 | 56.10 |
| EigenWorms | 64.58 | 44.20 | 97.85 | 99.33 | 83.00 | 90.33 | 72.16 | 86.28 | 62.80 | 74.68 | 81.93 | 76.62 | 78.17 | 83.00 | 45.45 | 98.68 | 89.31 |
| Epilepsy | 96.30 | 67.03 | 97.37 | 99.64 | 97.34 | 98.38 | 99.93 | 99.08 | 99.83 | 98.74 | 99.86 | 99.83 | 100.00 | 96.09 | 98.16 | 98.65 | 98.55 |
| EthanolConcentration | 30.15 | 30.68 | 29.87 | 48.64 | 34.06 | 72.89 | 60.18 | 44.68 | 39.62 | 82.36 | 49.16 | 45.42 | 80.68 | 28.99 | 28.62 | 27.92 | 43.73 |
| ERing | 92.91 | 91.42 | 92.89 | 96.89 | 91.98 | 95.65 | 93.19 | 98.05 | 84.48 | 84.28 | 82.44 | 89.84 | 94.26 | 89.46 | 87.19 | 92.10 | 84.10 |
| FaceDetection | 53.28 | 51.53 | – | 68.89 | 55.06 | 69.17 | 62.97 | 69.42 | 52.32 | 69.76 | 51.17 | 68.95 | 69.17 | 52.87 | 53.13 | 77.24 | 69.97 |
| FingerMovements | 54.17 | 55.50 | 54.93 | 54.77 | 54.43 | 53.90 | 55.53 | 55.27 | 51.03 | 53.40 | 52.10 | 53.17 | 53.77 | 51.33 | 54.70 | 56.13 | 65.00 |
| HandMovementDirection | 30.32 | 26.67 | 30.72 | 38.02 | 32.07 | 35.13 | 35.23 | 44.59 | 28.87 | 34.95 | 28.24 | 48.51 | 37.79 | 32.34 | 35.32 | 42.39 | 46.00 |
| Handwriting | 61.21 | 34.33 | 60.55 | 51.85 | 36.06 | 52.21 | 54.04 | 56.67 | 49.09 | 28.77 | 18.27 | 36.42 | 50.41 | 32.95 | 59.78 | 65.74 | 57.10 |
| Heartbeat | 68.88 | 63.80 | 69.87 | 73.59 | 78.49 | 76.52 | 72.52 | 71.76 | 72.15 | 72.15 | 73.22 | 72.28 | 72.18 | 73.97 | 63.89 | 78.62 | 80.49 |
| Libras | 88.04 | 78.63 | 87.85 | 90.30 | 75.56 | 91.67 | 86.57 | 90.61 | 85.26 | 84.46 | 81.67 | 79.72 | 90.28 | 83.63 | 94.11 | 88.72 | 83.33 |
| LSST | 54.76 | 49.57 | 56.96 | 63.62 | 58.05 | 56.17 | 60.28 | 63.15 | 43.62 | 57.82 | 50.58 | 34.31 | 53.84 | 46.33 | 42.94 | 33.97 | 68.53 |
| MotorImagery | 56.10 | 49.63 | 50.37 | 52.17 | 53.80 | 51.80 | 53.00 | 53.13 | 52.37 | 50.83 | 49.83 | 53.80 | 52.17 | 45.37 | 49.77 | 51.17 | 61.90 |
| NATOPS | 82.04 | 76.07 | 81.48 | 87.13 | 82.37 | 84.41 | 86.43 | 88.54 | 82.48 | 84.35 | 80.59 | 77.72 | 82.85 | 90.30 | 97.11 | 96.63 | 98.89 |
| PenDigits | 99.28 | 99.22 | 99.27 | 98.68 | 91.27 | 98.97 | 97.14 | 99.56 | 95.61 | 97.70 | 87.47 | 94.11 | 97.19 | 93.65 | 99.64 | 99.68 | 98.10 |
| PEMS-SF | 77.05 | 80.23 | 78.73 | 99.85 | 91.27 | 99.86 | 97.15 | 85.63 | 96.57 | 98.40 | 98.98 | 96.76 | 97.98 | 79.21 | 81.95 | 82.83 | 94.80 |
| PhonemeSpectra | 15.39 | 10.18 | – | 25.86 | 15.27 | 32.87 | 30.86 | 28.35 | 19.43 | 30.62 | 26.78 | 14.52 | 32.87 | 22.17 | 15.39 | 36.74 | 29.00 |
| RacketSports | 85.64 | 81.69 | 85.79 | 89.56 | 87.79 | 89.30 | 88.73 | 92.79 | 88.90 | 88.09 | 84.17 | 88.29 | 90.64 | 85.81 | 91.23 | 91.69 | 92.10 |
| SelfRegulationSCP1 | 81.81 | 80.63 | 81.34 | 73.58 | 79.74 | 85.94 | 82.86 | 86.55 | 81.33 | 84.73 | 73.17 | 84.73 | 86.02 | 95.68 | 76.11 | 84.69 | 90.00 |
| SelfRegulationSCP2 | 54.09 | 48.48 | 52.43 | 49.52 | 50.62 | 48.87 | 49.61 | 51.35 | 50.62 | 51.63 | 50.28 | 50.62 | 51.67 | 56.05 | 50.24 | 52.04 | 62.00 |
| StandWalkJump | 22.00 | 35.78 | 25.56 | 34.67 | 38.44 | 45.11 | 42.00 | 45.56 | 36.89 | 44.00 | 34.00 | 33.33 | 40.67 | 35.11 | 30.89 | 42.00 | 67.00 |
| UWaveGestureLibrary | 92.28 | 87.58 | 91.51 | 90.39 | 89.59 | 92.42 | 91.32 | 94.43 | 86.13 | 87.03 | 71.11 | 85.05 | 91.31 | 89.59 | 88.35 | 91.23 | 91.00 |

Table 11: Detailed predictive performance comparison across 26 datasets in UEA (supporting results for Section 3.2 in the main paper).

Table 12: `TimeSliver` vs. 16 baselines on 26 UEA datasets, evaluated using Average Rank, Top-1 Count, and Top-3 Count. **Bold** indicates the best result, underlined indicates the second-best. ↑ and ↓ denote that higher and lower values are better, respectively.

| Type | Method | Average Rank ↓ | Top-1 Count ↑ | Top-3 Count ↑ |
|---|---|---|---|---|
| Distance-Based | DTW_D | 16.88 | 0 | 0 |
| | DTW_I | 18.65 | 0 | 0 |
| | DTW_A | 14.67 | 1 | 1 |
| Dictionary-Based | MUSE | 10.50 | 3 | 6 |
| | gRSF | 16.88 | 0 | 0 |
| | CIF | 11.31 | 1 | 3 |
| Feature-Based ML | MrSEQL | 12.88 | 0 | 0 |
| | ROCKET | 9.31 | 4 | 7 |
| Deep Learning | TapNet | 17.77 | 1 | 1 |
| | ResNet | 15.38 | 1 | 1 |
| | IncTime | 7.69 | 4 | 7 |
| | FCN | 9.92 | 1 | 5 |
| | TS2Vec | 16.88 | 0 | 0 |
| | TimesNet | 15.92 | 0 | 0 |
| | ShapeNet | 12.58 | 1 | 2 |
| | RLPAM | 10.23 | 4 | 10 |
| | ShapeConv | 11.19 | 1 | 4 |
| | SBM | 11.15 | 1 | 3 |
| | InterpGN | 7.08 | 3 | 7 |
| **Ours** | TimeSliver | **7.00** | **6** | **11** |

## F.2 SENSITIVITY

To satisfy the *sensitivity* axiom, any change or perturbation ($\delta_s$) to a temporal segment $\mathbf{x}_s = \mathbf{x}[t : t + m]$ that induces a change in the predicted logit $\hat{y}_c$ must yield a nonzero attribution score for that segment, i.e.,

$$\hat{y}_c(\mathbf{x}) \neq \hat{y}_c(\mathbf{x}_{[x_s \leftarrow x_s + \delta_s]}) \implies \phi_s \neq 0,$$

In `TimeSliver`, the attribution score computation (Equations 3 and 4) involves two key steps after training:

- **Sensitivity of $\hat{y}_c$ with respect to $P$.** We first estimate $g_{ij} = \frac{\partial \hat{y}_c}{\partial P_{ij}}$. Although gradient-based methods typically violate sensitivity (Shrikumar et al., 2017), `TimeSliver` preserves it because $\hat{y}_c$ depends *linearly* on $P$ (Equation 9). Thus, any perturbation in $P$ that affects $\hat{y}_c$ necessarily yields $g_{ij} \neq 0$.

- **Separating positive and negative contributions.** In the second step, we use $\sigma_{ij} = \text{sign}(w_{ij})$ together with the $Z_{ki}Q_{kj}$ terms in Equation 3 to isolate positively and negatively contributing segments via the ReLU operator. Consequently, `TimeSliver` satisfies sensitivity *partially*: $\phi^+$ is insensitive to negatively contributing segments (they are zeroed), and $\phi^-$ is insensitive to positively contributing segments.

## F.3 SYMMETRY-PRESERVING

For a method to be symmetry-preserving, it must assign identical attribution scores ($\phi^+$ and $\phi^-$) to two identical temporal segments in the input sequence. To illustrate that `TimeSliver` satisfies this property, consider a univariate time-series instance $\mathbf{x} \in \mathbb{R}^{L \times 1}$ containing two identical segments $\mathbf{x}_s = \mathbf{x}[t_1 : t_1 + m]$ and $\mathbf{x}'_s = \mathbf{x}[t_2 : t_2 + m]$. As shown in Equation 3, the attribution score for a segment $\mathbf{x}_i$ primarily depends on $Z_i$ and $Q_i$. Section 2.2.2 establishes that $Z$ depends solely on the one-hot encoding $\mathcal{O}$ (Equation 1). Because identical raw segments yield identical one-hot encodings, we have $\mathcal{O}_s = \mathcal{O}'_s$. Likewise, the latent representation $\mathbf{q}_i$ is obtained from a 1D CNN applied at the

segment level; therefore, identical segments produce identical latent representations, i.e., $\mathbf{q}_s = \mathbf{q}'_s$. Consequently, both $Z$ and $Q$ are identical for the two segments. By Equation 3, this yields identical attribution scores $\zeta^+$ and $\zeta^-$, and therefore identical final attributions $\phi^+$ and $\phi^-$.

| Activation function | FreqSum | SeqComb-UV | SeqComb-MV | LOWVAR |
|---|---|---|---|---|
| ReLU (with max-scaling) | $\mathbf{0.94}_{\pm 0.05}$ | $\mathbf{0.97}_{\pm 0.03}$ | $\mathbf{0.94}_{\pm 0.01}$ | $\mathbf{0.99}_{\pm 0.04}$ |
| ReLU (w/o max-scaling) | $0.91_{\pm 0.06}$ | $0.94_{\pm 0.05}$ | $\mathbf{0.94}_{\pm 0.05}$ | $\mathbf{0.99}_{\pm 0.01}$ |
| Sigmoid | $0.91_{\pm 0.07}$ | $0.77_{\pm 0.18}$ | $0.75_{\pm 0.19}$ | $\mathbf{0.99}_{\pm 0.02}$ |
| Tanh | $0.91_{\pm 0.06}$ | $0.75_{\pm 0.20}$ | $0.70_{\pm 0.17}$ | $\mathbf{0.99}_{\pm 0.05}$ |
| No activation | $0.91_{\pm 0.07}$ | $0.75_{\pm 0.20}$ | $0.70_{\pm 0.17}$ | $\mathbf{0.99}_{\pm 0.04}$ |

Table 13: Comparison of activation functions across different tasks.

## G  QUALITATIVE ANALYSIS

Figure 9 presents qualitative results comparing the positive attribution scores computed by `TimeSliver` with the ground truth attribution scores for all synthetic datasets. The strong alignment between `TimeSliver`'s attribution scores and the ground truth demonstrates that high attribution scores accurately identify the truly important time points in $\mathbf{x}_i$.

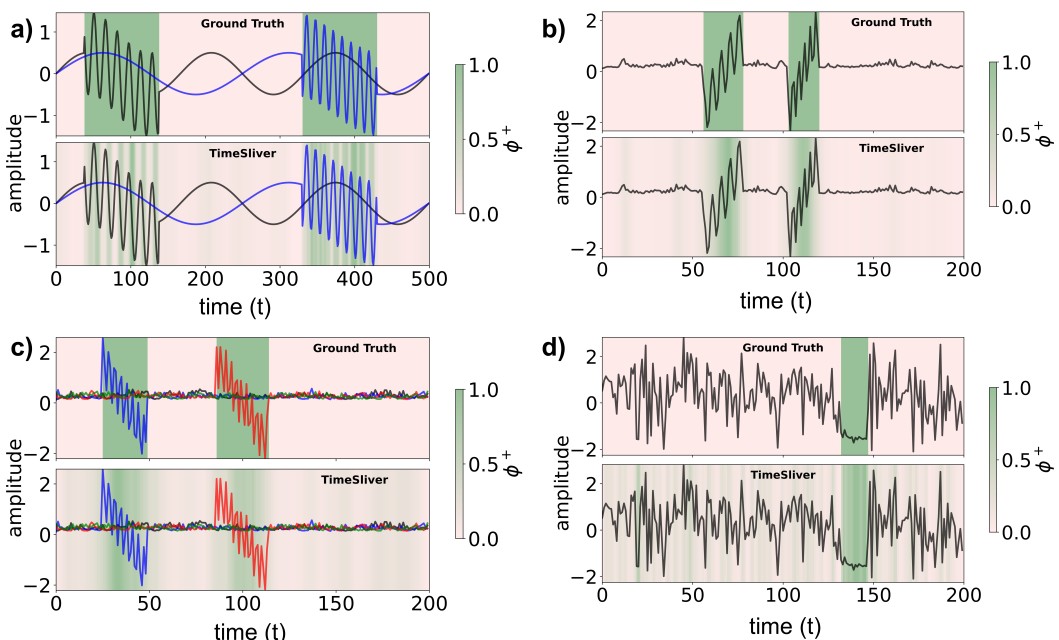

Figure 9: Qualitative comparison of temporal attribution scores obtained from `TimeSliver` with ground truth importance scores on the a) FreqSum, b) SeqComb-UV, c) SeqComb-MV, and d) LOWVAR datasets.

