# OpenReview forum: "TIMESLIVER : SYMBOLIC-LINEAR DECOMPOSITION FOR EXPLAINABLE TIME SERIES CLASSIFICATION"
_ICLR.cc/2026/Conference — ICLR 2026 Poster_

### Official Review · Reviewer_5bwY · 2025-10-30

**Soundness:** 3
**Presentation:** 3
**Contribution:** 2
**Rating:** 6
**Confidence:** 3

**Summary:**

This paper proposes a framework, TimeSliver, to generate faithful measures of temporal feature importance. The framework constructs temporal interpretations by utilizing a combination of temporal segments and a symbolic representation. The effectiveness of this approach is demonstrated through experiments on both synthetic and real-world datasets, where it is compared against several baseline methods.

**Strengths:**

•	The proposed framework's combination of temporal segment representation and symbolic composition-based representation is a novel approach for constructing measures of temporal importance.

•	The division of temporal attributions into positive and negative contributions is a valuable feature, as it provides insight into the directionality of a feature's impact on the model's output.

•	The experiments are thorough, demonstrating that TimeSliver can achieve significant improvements over various baselines on both synthetic and real-world datasets while maintaining high predictive performance.

**Weaknesses:**

•   **Clarity of Methodological Integration**: The paper incorporates advanced signal processing concepts, such as the Short-Time Fourier Transform (STFT). However, the specific mechanism by which the output of the STFT is integrated into the model's architecture and used for prediction is not sufficiently detailed.

•	**Rationale for Design Choices**: Several core methodological definitions and design choices could be further justified. For example, the formal definition of "temporal attribution-based explainability" is somewhat abstract, and the rationale for specific formulations, such as the interaction matrix P and the use of the ReLU function for attributions, is not fully explained.

**Questions:**

**Main Concerns:**

1.	**Definition of Explainability (Definition 2.1)**: The concept of "Temporal Attribution-Based Explainability" is defined by the "model’s ability" to assign importance scores. This definition is somewhat ambiguous. Could the authors provide a more formal or precise definition of "ability" in this context and clarify how it is quantitatively measured?
2.	**Interaction Matrix Formulation (Sec 2.2.3)**: The matrix $P$ is defined as a linear weighted summation of all temporal segments, yet it is purported to capture interactions. Could you elaborate on how this linear formulation is sufficient to capture pairwise or higher-order interactions, which are typically non-linear?
3.	**Use of ReLU for Attributions (Equation 3)**: Equation (3) utilizes the ReLU function to normalize and separate the positive and negative attributions. Could you provide the justification for choosing ReLU for this task over other potential functions? What is the key insight behind this specific formulation for deriving positive and negative contributions?
4.	**STFT (Line 202)**: The framework computes the Short-Time Fourier Transform. Could the authors please clarify how the resulting frequency-domain features are subsequently used in the prediction model?
5.  **Qualitative Analysis of Experiments**: While the paper presents a framework for interpretability evaluation (e.g., via masking-based metrics), it lacks intuitive visualization and deeper qualitative analysis. The authors do not show how the model explains individual samples — for example, which time points or variables are most critical for a specific prediction, and whether these insights align with domain expertise.

**Minor Concerns:**

1.	**Notation (Lines 131-144)**:  input vectors ($x_i$) in boldface in lines 131-134, 137, 139, and 144.

2.	**Attribution Function (Line 122)**: The notation seems to imply that $\alpha_i = f_{att}(\hat{y}_i) $ . This suggests the attribution is a function of the model's prediction, which may not be the intended meaning. Please clarify this relationship.
3.	**Equation (3)**: The expression for the positive attribution A⁺ appears to be missing the arguments $g_{ij}$ and $\sigma_{ij}$.

---

> ### Author Response · Authors · 2025-11-21
> **Response to Reviewer 5bwY [1/3]**
>
> We sincerely thank the reviewer for the thoughtful and constructive feedback, and for recognizing the novelty of our symbolic-temporal framework, the value of distinguishing positive and negative attributions, and the thoroughness of our experimental evaluation. Below, we address the reviewer's concerns (Weakness denoted as W, and Questions as Q) point-by-point to strengthen the paper.
>
>
> ## Questions
>
> > Q1. Definition of Explainability (Definition 2.1): The concept of "Temporal Attribution-Based Explainability" is defined by the "model’s ability" to assign importance scores. This definition is somewhat ambiguous. Could the authors provide a more formal or precise definition of "ability" in this context and clarify how it is quantitatively measured?
>
> In Definition 2.1, "model's ability" refers to how accurately the model's temporal attribution scores align with ground-truth importance scores. To quantitatively evaluate this alignment, we employ AUPRC for synthetic datasets (where ground truth is available) and $I(100)$ and $I(20)$ for real-world datasets, as detailed in **Section 2.2.5**. **Tables 1, 2, and 3** present comprehensive comparisons demonstrating that TimeSliver consistently outperforms baseline methods across these metrics. Based on the reviewer's valuable feedback, we have revised **Definition 2.1** and the **Problem Statement** to provide greater clarity regarding temporal attribution-based explainability.
>
> ---
>
> > W2, Q2. Interaction Matrix Formulation (Sec 2.2.3): The matrix
>  is defined as a linear weighted summation of all temporal segments, yet it is purported to capture interactions. Could you elaborate on how this linear formulation is sufficient to capture pairwise or higher-order interactions, which are typically non-linear?
>
>
> $\texttt{TimeSliver}$ captures pairwise and higher-order interactions through three complementary mechanisms:
>
> - **Non-linear latent representations (Module I, Section 2.2.1)**: Learning latent space representations of temporal segments using non-linear 1D CNN layers $g(\theta_q)$, which encode complex patterns within each segment
>
> - **Positional encoding for sequential dependencies (Module III, Section 2.2.3)**: Adding sinusoidal positional encoding to $\mathbf{x}_i$ before the 1D CNN to capture explicit temporal ordering when sequence matters
>
> - **Multi-scale temporal modeling (Module III, Section 2.2.3)**: Computing $\boldsymbol{P}$ across multiple segment sizes $\{m_1, m_2, \ldots, m_{|m|}\}$ and stacking them into $\boldsymbol{\mathcal{R}} \in \mathbb{R}^{(n \cdot v) \times q \times |m|}$ to capture interactions at different temporal scales
>
> **Empirical validation**: To demonstrate $\texttt{TimeSliver}$'s ability to capture temporally disjoint yet jointly informative patterns, we constructed a synthetic dataset with multiplicative interactions between distant segments (Appendix D.3) and evaluated its predictive performance against multiple architectures. As shown in Appendix D.3, $\texttt{TimeSliver}$ achieves performance within 1% of the baselines, confirming its effectiveness in modeling temporally disjoint interactions despite the linear aggregation in Equation 2.
>
> ---

---

> ### Author Response · Authors · 2025-11-21
> **Response to Reviewer 5bwY [2/3]**
>
> > W2, Q3. Use of ReLU for Attributions (Equation 3): Equation (3) utilizes the ReLU function to normalize and separate the positive and negative attributions. Could you provide the justification for choosing ReLU for this task over other potential functions? What is the key insight behind this specific formulation for deriving positive and negative contributions?
>
> We chose ReLU over other activation functions for computing positive and negative attribution scores due to three key reasons:
>
> 1. **Clear directional separation**: ReLU cleanly distinguishes between positive and negative attributing segments
> 2. **Magnitude sensitivity**: ReLU preserves the relative importance of contributions without saturation (details in Appendix F.2)
> 3. **Superior empirical performance**: ReLU achieves significantly higher AUPRC scores compared to Tanh and Sigmoid, as demonstrated in **Table 13 (Appendix)**. We have added this quantitative result in the form of a table at the end of this response.
>
> ### Example
>
> Consider $P_{ij} = \sum_{k=0}^{4} Z_{ki}Q_{kj} = -1 + 0 + 0.5 + 10 + 15$ (from Equation 2).
>
> The positive attribution scores ($\phi^+_k$) for five temporal segments, computed **without normalization** for clarity, are:
>
> | Segment contributions | -1 | 0 | 0.5 | 10 | 15 |
> |----------------------|-----|-----|------|-----|-----|
> | **Sigmoid** | 0.27 | 0.5 | 0.62 | 1.0 | 1.0 |
> | **Tanh** | -0.76 | 0 | 0.46 | 1.0 | 1.0 |
> | **ReLU** | 0 | 0 | 0.5 | 10 | 15 |
>
> **Key observations:**
> - **Sigmoid** fails to distinguish direction—the negatively contributing segment (contribution = -1) receives a positive score (0.27)—and saturates at 1.0 for large values, losing magnitude information
> - **Tanh** shows direction but saturates at 1.0 for both segments with contributions of 10 and 15, making them indistinguishable despite a 1.5× difference in importance
> - **ReLU** correctly assigns zero attribution to the opposing segment (contribution = -1), zeros out neutral segments, and preserves the magnitude relationships (10 vs. 15), enabling accurate identification of the most important time points
>
>
>
> | Activation function | FreqSum | SeqComb-UV | SeqComb-MV | LOWVAR |
> |---------------------|---------|------------|------------|--------|
> | ReLU (with max-scaling) | **0.94**±0.05 | **0.97**±0.03 | **0.94**±0.01 | **0.99**±0.04 |
> | ReLU (w/o max-scaling) | 0.91±0.06 | 0.94±0.05 | **0.94**±0.05 | **0.99**±0.01 |
> | Sigmoid | 0.91±0.07 | 0.77±0.18 | 0.75±0.19 | **0.99**±0.02 |
> | Tanh | 0.91±0.06 | 0.75±0.20 | 0.70±0.17 | **0.99**±0.05 |
> | No activation | 0.91±0.07 | 0.75±0.20 | 0.70±0.17 | **0.99**±0.04 |
>
> ---
>
>
> > W1, Q4. STFT (Line 202): The framework computes the Short-Time Fourier Transform. Could the authors please clarify how the resulting frequency-domain features are subsequently used in the prediction model?
>
> We want to clarify that we do not use any STFT features in the $\texttt{TimeSliver}$ framework. We use the cross-representation matrix $\boldsymbol{P} = \boldsymbol{Z}^\top \boldsymbol{Q}$ to predict the target labels. The **Remark (Approximate Structural Analogy of $\boldsymbol{Z}$ with Spectral Representation)** in Section 2.2.2 serves as a juxtaposition of our composition matrix $\boldsymbol{Z}$ with time-frequency transformations like STFT to provide intuition into our design components.
>
>
> ---
>
>
> > Q5. Qualitative Analysis of Experiments: While the paper presents a framework for interpretability evaluation (e.g., via masking-based metrics), it lacks intuitive visualization and deeper qualitative analysis. The authors do not show how the model explains individual samples — for example, which time points or variables are most critical for a specific prediction, and whether these insights align with domain expertise.
>
> Based on the reviewer's feedback, we have included qualitative results comparing the positive attribution scores computed by $\texttt{TimeSliver}$ with the ground truth attribution scores for all synthetic datasets. These results are presented in **Figure 9** in **Appendix G**. The strong alignment between $\texttt{TimeSliver}$'s attribution scores and the ground truth demonstrates that high attribution scores correspond to the truly important time points in $\mathbf{x}_i$.
>
> In the current work, our qualitative analysis is limited to synthetic datasets, as extending it to real-world datasets would require human-in-the-loop validation. While this is beyond the scope of our current study, it represents a natural and important direction for future work.
>
>
> ---

---

> ### Author Response · Authors · 2025-11-21
> **Response to Reviewer 5bwY [3/3]**
>
> ### Minor questions
>
> > Q6. Notation (Lines 131-144): input vectors ($x_i$) in boldface in lines 131-134, 137, 139, and 144.
>
> Based on the reviewers feedback, we have consistently changed $x_i$ to $\mathbf{x_i}$.
>
> ---
>
> > Q7. Attribution Function (Line 122): The notation seems to imply that $\alpha_i = f_{att}(\hat{y}_i)$. This suggests the attribution is a function of the model's prediction, which may not be the intended meaning. Please clarify this relationship.
>
> We have revised the wording and notation in **Section 2.1**, with particular attention to the **Problem Statement**, to enhance clarity. Importantly, we have reused our notations of $\{\phi^+, \phi^-\}$ instead of introducing a new notation $\alpha_i$, and updated the dependence of $f_{\mathrm{att}}(\cdot)$ on the input features and the predicted logits clearly in the **Problem Statement**.
>
>
> ---
>
> > Equation (3): The expression for the positive attribution $\zeta$ appears to be missing the arguments $g_{ij}$ and $\sigma_{ij}$.
>
> We have included the suggested change in Equation 3.
>
> ---

---

> > ### Comment · Reviewer_5bwY · 2025-11-27
> > **reply to authors**
> >
> > Thanks for the authors' response.  I will maintain my positvie ratings.

---

### Official Review · Reviewer_A8un · 2025-10-31

**Soundness:** 3
**Presentation:** 3
**Contribution:** 2
**Rating:** 6
**Confidence:** 3

**Summary:**

The paper proposes a new framework named TimeSliver, targeting faithful explainability and strong performance. The authors point out that traditional deep-learning models often suffer from the black-box nature of neural networks. Furthermore, existing post-hoc methodologies (e.g., Grad-CAM) or approaches utilizing attention weights suffer from an "unfaithfulness" issue, failing to address the model’s reasoning. TimeSliver attempts to solve this by decomposing the input into symbolic and latent representations of time-series data.

To elaborate, the framework processes the input through two parallel modules. The first module learns a latent representation using a 1D convolution operator, targeting the corresponding temporal segments. The second module creates a symbolic composition matrix by discretizing the raw input into categorical bins and applying average pooling over the fixed segment windows to capture the normalized frequency of each symbol within each segment. The key module, Module 3, computes a global representation by learning the ‘symbolic-linear’ representation (a cross-representation matrix) composed of the results from Modules 1 and 2. This structure allows TimeSliver to decompose the final prediction and assign positive/negative attribution scores to every temporal segment.

Experiments were conducted on three real-world applications and four synthetic datasets, comparing TimeSliver with nine baseline methods. Additionally, its performance was evaluated on 26 multivariate time-series classification tasks.

**Strengths:**

The paper's primary strength lies in its ability to achieve consistently better performance than existing methodologies in most tested cases. It is impressive that this strong performance is achieved using a methodology that is relatively simple and computationally efficient. This simplicity is a significant advantage, as the model is designed with a highly intuitive intention, ensuring that the authors' original goal is well-aligned with the final model architecture and its effective results.

**Weaknesses:**

The authors provide compelling evidence that the symbolic component (Module 2) is essential for the model's explainability goal; replacing the symbolic matrix $Z$ with a non-symbolic projection $X_{proj}$ significantly degrades explainability metrics (AUPRC), as shown in Figure 3 and Table 9 . Given that the paper's main objective is not necessarily maximizing predictive performance, it would nonetheless be beneficial to understand the full impact of this substitution. The paper does not appear to report the predictive accuracy results for this specific ablation study, accuracy results without module 2. Providing this information would offer a more complete characterization of the framework, clarifying whether the symbolic module also contributes to predictive accuracy or if its role is only focused on enabling the model's core explainability goal.

**Questions:**

The paper clearly demonstrates that replacing the symbolic matrix $Z$ with $X_{proj}$ significantly harms explainability metrics (AUPRC), as shown in Table 9 1. What was the impact on predictive accuracy in that same ablation experiment?
More broadly, what is the exact influence of Module 2 on the model's final predictive performance and the training of the other modules?

---

> ### Author Response · Authors · 2025-11-21
> **Response to Reviewer A8un**
>
> We sincerely thank the reviewer for the comprehensive and insightful summary of our work, and for recognizing TimeSliver's consistent performance improvements, computational efficiency, and intuitive design that aligns methodology with explainability goals. Below, we address the reviewer's questions and concerns.
>
> ## Weaknesses
>
> > W1. The authors provide compelling evidence that the symbolic component (Module 2) is essential for the model's explainability goal; replacing the symbolic matrix $Z$ with a non-symbolic projection $X_{proj}$ significantly degrades explainability metrics (AUPRC), as shown in Figure 3 and Table 9 . Given that the paper's main objective is not necessarily maximizing predictive performance, it would nonetheless be beneficial to understand the full impact of this substitution. The paper does not appear to report the predictive accuracy results for this specific ablation study, accuracy results without module 2. Providing this information would offer a more complete characterization of the framework, clarifying whether the symbolic module also contributes to predictive accuracy or if its role is only focused on enabling the model's core explainability goal.
>
> Based on reviewer's feedback, we have revised **Figure 3 (Section 2.2.4)** to illustrate the effect of replacing symbolic discretization **(Module II, Section 2.2.2)** with non-symbolic projection $X_{proj}$ on accuracy. The updated figure demonstrates that this substitution substantially impairs explainability (17% AUPRC reduction) without affecting predictive performance (accuracy), confirming the necessity of symbolic encoding for faithful temporal attribution.
>
> **Table 9 (Appendix D.2)** has been expanded to include predictability results. The updated table confirms that while symbolic discretization is critical (its removal degrades AUPRC by 38.7%), the choice of discretization method have minimal impact. Both ABBA and SFA achieve equivalent explainability and predictability to our binning approach, with accuracy varying by less than 1% across all methods.
>
>
> ## Questions
>
> > Q1. The paper clearly demonstrates that replacing the symbolic matrix
>  with $Z$ with $X_{proj}$ significantly harms explainability metrics (AUPRC), as shown in Table 9 1. What was the impact on predictive accuracy in that same ablation experiment? More broadly, what is the exact influence of Module 2 on the model's final predictive performance and the training of the other modules?
>
> We have now revised our **Figure 3 (Section 2.2.4)** and **Table 9 (Appendix D.2)** to illustrate the effect of replacing symbolic discretization **(Module II, Section 2.2.2)** with non-symbolic projection $X_{proj}$ on the predictive performance.
>
> ---

---

> > ### Author Response · Authors · 2025-11-28
> >
> > We wanted to reach out to kindly ask if our response has answered your question on the effect of the symbolic matrix $Z$ on the predictive capability of TimeSliver. Thank you for your time.

---

### Official Review · Reviewer_Cxgd · 2025-10-31

**Soundness:** 2
**Presentation:** 2
**Contribution:** 3
**Rating:** 4
**Confidence:** 4

**Summary:**

The paper introduces TimeSliver, and explainable time series model aim to explain both positive and negative contributions in terms of which time points are most important for the prediction.

**Strengths:**

- The writing of the paper is clear for some sections (Section 2.2.1 to 2.2.3)
- The methods are intuitive and understandable (Section 2.2.1 to 2.2.3)
- The evaluation methods of the results are pretty sound.

**Weaknesses:**

- There are some notations and logic disconnect in the method section (Section 2.2.3 to 2.2.4). Notations such as $f_{cls}$ and $f_{att}$ should be used in section 2.2.4 but they are not.
- Justifications and explanations of the formulae for the attributions are missing. The author instead only focuses on the justification of the scaling aspect on the formula.
- The definition of positive and negative contributions are not there. In the formula it seems to suggest one thing but in the evaluation it seems to suggest another.
- The evaluation results are not super impressive. This is okay if the method section is strong and sound.

More explanations are in the "Questions" below.

**Questions:**

Question

- Line 121. It is better to say "comprising two components - predictions and attribution". Note that this notation $f_{cls}$ and $f_{att}$ is never used again. Also by the formula, the domain of $f_{att}$ is $\{0, \ldots, C-1\}$, which is very weird. The $\alpha_i$ should depend on the input x and the network as well (as shown in Fig 1). So I think the notations have to be updated.

- In section 2.2.1, I am confused with the statement "partitions x_i into ... overlapping segments". If we are partitioning the time series, they would not overlap. Secondly, using a 1d convolutional operator with kernel size m, the resulting "output" would not be a "temporal segment", as a "temporal segment" is defined to be a subsequence of the original segment $x_i$. Convolving with a kernel would mostly not result in a subsequence of the original sequence. So I think we should change the wording here a little bit.

- There is a disconnect from Section 2.2.3 to Section 2.2.4. From what I understand from the introduction, this work TimeSliver, is not a post-hoc interpretation method, but a model with explanation. The author described how to get the matrices P's from the input time series x, but what I assumed here is that there is an extra model that treats P as the input features and output y. This is in reference to Figure 1, the "$f_{cls}$". However, in section 2.2.4, it says "Once the model is trained to predict y", it could mean the model is trained separately. So I think here, it is better to add a sentence here that "we are training a model $f_{cls}$ using $P$ as the features.

- Continuing with the above $f_{att}$ should be described that it takes the output of $f_{cls}$ and $P$ as the inputs and relate it to $\phi^+_k$ and $\phi^-_k$ in equation 4.

- For equation 3, what if the terms in $P_{ij}$ are all positive or negative (across $k$)? Then the denominator would be undefined?

- For equation 3, there does not seem to have any discussion on this definition. Although it was mentioned that more explanations are in Section A in the appendix, the explanations are only on the "scaling". Thus, if I am to interpret it mayself, $g_{ij}$ is the gradient of $\hat{y_c} ^p$ w.r.t. $P_{ij}$ . This means that it is the rate of change of the prediction when $P_{ij}$ changes. $g_{ij}$ is positive when the prediction for the class increases when $P_{ij}$ increases. But $P_{ij}$ is written as a sum of the $k$ terms, corresponds to the time k. Thus if the $k$ time point contribution to $P_{ij}$ is positive, then this $\zeta$ would become the "positive contribution". Multiplying by the denominator will have a scaling discussion as the text following this equation. So **does that mean that a positive contribution means how much the data time point $k$ would drive the prediction higher, while a negative contribution means how much the data time point $k$ would drive the prediction lower?** In the introduction and contributions, the author mentioned that TimeSliver provides positive and negative temporal attribution scores. Are there any definitions on what this actually means? From the formula, I think the answer is yes.

- Continuing with the above observation. If the answer is yes to the bolded question, then negative contributions are also good predictors as they would drive the prediction of the class lower. This could help the model performance if the ground truth class is not that class. Thus I am not following the discussion on paragraph in Lin 311 to 315. It seems like the authors are claiming that "negative contributions" correspond to "noisy time points", instead of "drivers to lower predictions".

- Table 4 is not referenced anywhere. I think it should be referenced from Section 3.2 in the experiment.

---

> ### Author Response · Authors · 2025-11-21
> **Response to Reviewer Cxgd [1/2]**
>
> Thank you for your time and constructive feedback. We appreciate your recognition of our method's solid intuition, clarity, and thorough experimental design. Below, we address your questions and comments.
>
> ## Weakness and Questions
>
> > Q1 : Line 121. It is better to say "comprising two components - predictions and attribution". Note that this notation $f_{cls}$ and $f_{att}$ is never used again. Also by the formula, the domain of $f_{att}$ is 0,...,C-1, which is very weird. The $\alpha_i$ should depend on the input x and the network as well (as shown in Fig 1). So I think the notations have to be updated.
>
> We have revised our notations for clarity and summarize them below:
>
> * $\mathbf{h}(\cdot; \theta_{\mathrm{q}})$ denotes the function that maps the input $\mathbf{x}_i$ to the latent representation $\boldsymbol{Q}$.
> * $\mathbf{g}(\cdot; \theta_{\mathrm{c}})$ denotes the function that maps the representation $\boldsymbol{P}$ to the predicted logits $\hat{y}_i$.
> * $f(\mathbf{x}_i; \theta_{\mathrm{q}}, \theta_{\mathrm{c}})$ denotes the overall predictive model.
> * $f_{\mathrm{att}}$ denotes the non-parametric transformation that generates temporal attribution scores $\phi$.
>
> We have updated our **Problem Statement, Figure 1, and Section 2** accordingly to ensure consistency.
>
> ---
> > Q2: In section 2.2.1, I am confused with the statement "partitions x_i into ... overlapping segments". If we are partitioning the time series, they would not overlap. Secondly, using a 1d convolutional operator with kernel size m, the resulting "output" would not be a "temporal segment", as a $x_i$ "temporal segment" is defined to be a subsequence of the original segment. Convolving with a kernel would mostly not result in a subsequence of the original sequence. So I think we should change the wording here a little bit.
>
> We chose overlapping segments to improve the temporal resolution. For instance, if we have a sequence length of $L=10$ and wish to generate segments of length $m=5$ from this original time series, we can obtain only 2 non-overlapping segments. However, if we overlap the segments such that subsequent segments start one time-step after the start-time of the first segment, we can obtain $\kappa = L - m + 1 = 6$ such segments and improve the temporal resolution of the final output. Thus, we use overlapping segments. Based on your feedback, we have updated our choice of word from ``partition`` to ``convert``.
>
> For the second issue, indeed conversion using a convolutional layer will not provide the exact input, and that is an intentional design choice to obtain a high-dimensional representation of the local subsequence of the original time series, as reflected in the title of *Section 2.2.1 - Module I: Latent Representation of Temporal Segments*. Based on your feedback, we have improved the writing of this section by explicitly stating -- *Given a multivariate time-series input $\mathbf{x}_i \in \mathbb{R}^{L \times v}$, this module converts $\mathbf{x}_i$ into $\kappa = L - m + 1$ overlapping latent representations. These representations are obtained using a 1D convolutional operator with kernel size $m$ and stride $1$ resulting in $\kappa$ sequences each of sequence length of $m$. Each segment captures a localized temporal context within the time series.*
>
>
> ---
> >Q3: Disconnect from Section 2.2.3 to Section 2.2.4
>
> >a) Missing explicit training of $f_{\mathrm{cls}}$:}
> Clarify that the linear classification model $f_{\mathrm{cls}}$ is trained on $\boldsymbol{P}$ as input features to predict $y$.
>
> >b) Unclear specification of $f_{\mathrm{att}}$:}
> Explicitly describe that $f_{\mathrm{att}}$ takes $\boldsymbol{P}$ and the outputs of $f_{\mathrm{cls}}$ (gradients and logits) as inputs to compute the attribution scores $\phi_k^+$ and $\phi_k^-$ in Equation 4.
>
> a. We have explicitly described the training of TimeSliver in **Section 2.2.3**. We have updated and consistently integrated these notations across the paper and **Figure 1**.
>
> b. In **Section 2.2.4**, we clearly describe how we leverage the optimized parameters $(\theta*_{\mathrm{q}}, \theta*_{\mathrm{c}})$, to compute the attribution scores using the non-parametric function $f_{\mathrm{att}}$. Since $\boldsymbol{P}$ and $\boldsymbol{Q}$ are derived from the parameters $\theta_{\mathrm{q}}$, and $\hat{y}$ is the predicted logit from parameters $\theta_{\mathrm{c}}$ for input an input $x$, our notation explicitly captures this dependency in **Sections 2.2.4 and 2.1 (Problem Statement)**.

---

> ### Author Response · Authors · 2025-11-21
> **Response to Reviewer Cxgd [2/2]**
>
> > Q4: For equation 3, what if the terms in $P_{ij}$ are all positive or negative (across $k$)? Then the denominator would be undefined?
>
>
> Your observation is correct: during our implementation we have used a small $\epsilon = 1\text{e}{-18}$ for numerical stability (please refer to our code file, $\texttt{timesliver.py}$ line 286 in the supplementary materials). We have now explicitly indicated this in **Section 2.2.4** of our paper as well.
>
> ---
> > Q5: Further explainations of Equation 3 beyond Appendix A :
>
> > a). **Does it mean that a positive contribution means how much the data time point $k$ would drive the prediction higher, while a negative contribution means how much the data time point would drive the prediction lower?**
>
>
> > b). In the introduction and contributions, the author mentioned that TimeSliver provides positive and negative temporal attribution scores. Are there any definitions on what this actually means? From the formula, I think the answer is yes.
>
> > c). Continuing with the above observation. If the answer is yes to the bolded question, then negative contributions are also good predictors as they would drive the prediction of the class lower. This could help the model performance if the ground truth class is not that class. Thus I am not following the discussion on paragraph in Lin 311 to 315. It seems like the authors are claiming that "negative contributions" correspond to "noisy time points", instead of "drivers to lower predictions".
>
> a, b). Yes, your understanding is correct. We have explicitly indicated the positive and negative attribution score definitions in **Definition 2.1 of the paper**.
>
> c). We want to clarify that the gradient $|g_{ij}|$ is obtained for the logit corresponding to the model's predicted class label. In this case, it is desirable to increase $\hat{y}_c^p$. Thus, the time-points that increase $\hat{y}_c^p$ positively influence the model's predictions and hence have higher positive attribution scores ($\phi_k^+$). Conversely, time-points that drive this value lower make the output less likely to be the predicted class label (for instance, when we take the softmax of all the logits to assign class labels, a lower value for $\hat{y}_c^p$ decreases the model's confidence in producing this class label), which means those time-points are detrimental to the model's prediction and are assigned higher negative attribution scores ($\phi_k^-$).
>
> Intuitively, masking such negatively influencing time-points should either improve the model's predictions or keep them unaffected. To assess this, we design the experiment in Table 3, where we mask the top 2\% and 5\% of the most negative time-segments. Empirically, we observe that especially on the EEG dataset, removing such negatively attributed time-segments yields an average prediction improvement of 4\%.
>
> Regarding whether cases of model misprediction: $\texttt{TimeSliver}$ follows similar convention as prior works in explainability [1, 2, 3] that are designed to explain the model's predictions regardless of their correctness. Moreover, during inference, we do not have access to the ground truth label, so incorporating interpretation based on the model's mispredictions is not practical.
>
>
> > Q6: Table 4 is not referenced anywhere. I think it should be referenced from Section 3.2 in the experiment.
>
> Thank you for pointing this out, we had earlier referenced it only in the **Appendix B.4**. We have now referred and highlighted it in **Section 3.2**.
>
> ---
> > Q7: The evaluation results are not super impressive. This is okay if the method section is strong and sound.
>
> We have constructed studies to specifically evaluate our method on real-world datasets by adopting masking-based performance evaluation. **Tables 2 and 3** show consistent improvements over baselines. We also evaluate our method against synthetic datasets where the ground truth importance is known and our method shows an average of **18\% improvement**.
>
> Additionally, we have refined the methodology section's writing based on your and other reviewers' feedback. We are happy to incorporate other updates to improve the clarity of our approach.
>
>
> [1] Integrated Gradients, ICLR 2018
>
> [2] Timex++, ICLR 2024
>
> [3] Grad-CAM, ICCV, 2017

---

> > ### Comment · Reviewer_Cxgd · 2025-11-24
> >
> > Thanks for addressing my concerns. I am satisfied with most of the concerns except for the negative attribution one.
> > - (Minor) line 277 $g(\cdot ; \theta_q)$ vs $g(\theta_q)$ (h similar) in Line 155 and Fig 1. I think we should use $g(\cdot ; \theta_q)$ (There is also a weird linebreak in Line 155)
> >
> > The main point I would like to discuss is the positive and negative attributions. In the rebuttal you mentioned
> >
> > > We want to clarify that the gradient is obtained for the logit corresponding to the model's predicted class label.
> >
> > While in line 291 it was
> >
> > > Let $\hat{y}_c$ denote the logit output corresponding to the class label of the input $x_c$.
> >
> > The line (291) is stating that $\hat{y}_c$ is the logit of the ground truth class while the line in the rebuttal is stating that it is the logit of the predicted class. Which is it?
> >
> > I completely missed this in my last review because calculating attribution should not require ground truths. Thus I think Line 291 is incorrect (while the line in the rebuttal is correct, as well as in line 127). So please update this.
> >
> > However, this would raise a more serious question. As $\hat{y}_c$ is the predicted class logit, the claim on "it is desirable to increase $\hat{y}_c$" is not always correct, as the model can make mistake. Thus it may not be true that the following statement holds.
> >
> > > Intuitively, masking such negatively influencing time-points should either improve the model's predictions or keep them unaffected.
> >
> > Masking negatively influencing time points would make the model more confident in their current predictions, which may or may not be a good thing. i.e. making the model more confident may increase or decrease the model's performance. For example,
> >
> > Case 1: The model is predicting the correct class, and feature A is having a negative attribution. In this case, that means that despite feature A, the model is still predicting the correct class, and thus removing feature A will keep the current prediction. (But removing feature A will bias the model to ignore the potential negative driver, which makes the model overconfident.)
> >
> > Case 2: The model is predicting the incorrect class, and feature A is having a negative attribution to the incorrect class. This means that it is possible that feature A will "drive" the model to predict a class that is different from the current predicted class. Removing feature A will make the model more confident in predicting the wrong class.
> >
> > A well-trained model should have more case 1s than case 2s. But no matter what, removing feature with negative attribution is not a good thing - thus negative attributions should not be treated as "noisy time points".
> >
> > > Regarding whether cases of model misprediction...
> >
> > I completely agree with this paragraph. We should explain the model's prediction, not explain the ground truth label. However, as the **evaluation of this explanation (the negative attributions) are done using the performance, which requires the ground truth label, it is the notion of "negative attributions being noisy" that bothers me**.
> >
> > So for suggestion, I would rather replace Table 3 with the table that shows that the magnitude of the increase of the "logits" (or the "probability") of the predicted class when negative attributions are masked. (A more direct way of showcasing the negative attributions)

---

> > > ### Author Response · Authors · 2025-11-26
> > >
> > > Thank you for thoughtful response. We have made the following changes in the updated manuscript based on your feedback:
> > >
> > > * Updated all references from $\boldsymbol{g}(\theta_q)$ to $\boldsymbol{g}(x_i; \theta_q)$ or $\boldsymbol{g}(\cdot; \theta_q)$.
> > >
> > > * Explicitly clarified that the instances of $\hat{y}_c$ refer to the predicted class logits.
> > >
> > > * Updated **Table 3** with the performance change with respect to the raw logits and introduced the new metric **$\Delta\hat{y}_c(u^-)$** in Section 2.2.5 to quantify this change. We have consistently revised our results in **Section 3.1** accordingly.
> > >
> > > > Regarding the negative attribution scores.
> > >
> > > * We are aligned with your observations—(1) that a well-trained model should have more cases of correct predictions, and (2) that explanations should be conducted based on the model’s predictions. These are our key motivations for developing $\texttt{TimeSliver}$: to design an explainable method with competitive predictive abilities. Accordingly, we derive the importance of the time points based on the predicted logits.
> > >
> > > * Based on your feedback, we now report the average change in the logit corresponding to the predicted class in Table~3. The results remain consistent with previously observed trends. On EEG, $\texttt{TimeSliver}$ achieves a mean increase of 0.26 in the predicted logit after masking the top negatively attributing time points, an increase that is 60\% higher than the next best baseline. In the Audio and FordA datasets, the mean change in the logit is close to zero, indicating the absence of negatively attributing time points for the majority of samples. Additionally, the standard deviation of $\Delta\hat{y}_c(u^-)$ indicates that negatively attributing time points may be present or absent on a per-sample basis, and we report the mean change in the predicted-class logit across all samples.
> > >
> > >
> > > * We have removed all instances where we described negative attribution scores as ``noisy``, since this reasoning was based on empirical trends in the EEG dataset (Table 3 and Figure 4). And based on the strong predictive performance in Table 4 and Table 8, we indeed have significantly more correct than incorrect predictions; however, as you suggested, the interpretation that negative attributions are noisy may not hold generally. Therefore, we no longer include this statement.
> > >
> > > We look forward to hearing your feedback and discussing any remaining concerns. Thank you for your time.

---

> > > > ### Comment · Reviewer_Cxgd · 2025-11-26
> > > >
> > > > Thank you for your response! That addressed my concerns.
> > > >
> > > > I also appreciate the authors to not showing the equation numbers to the new changes for the purpose of reviews - but they should be added in the camera ready version.

---

> > > > > ### Author Response · Authors · 2025-11-26
> > > > > **Official Comment by Authors**
> > > > >
> > > > > Thank you for your prompt response and feedback. We are glad that we were able to resolve your concerns.

---

### Official Review · Reviewer_VCe7 · 2025-11-01

**Soundness:** 3
**Presentation:** 3
**Contribution:** 3
**Rating:** 4
**Confidence:** 3

**Summary:**

The paper proposes TimeSliver, a deep learning framework that combines raw time series with symbolic representations to provide temporal attribution scores for explainable time series classification. The method achieves competitive predictive performance while offering interpretable insights into which time segments influence predictions.

**Strengths:**

1. Novel architectural design: The combination of symbolic abstraction with raw time series through linear composition ($P = Z^T Q$) is creative and well-motivated. The structural analogy to STFT provides good intuition.
2. Strong experimental validation: The paper demonstrates 11-18% improvement over baselines on synthetic datasets and maintains competitive performance on 26 UEA benchmark datasets while providing explainability.
2. Scale-invariance property: The theoretical justification for why symbolic representation yields scale-invariant attributions (Section A) is valuable and validated empirically.

**Weaknesses:**

1. baseline comparisons: Using computer vision methods (Grad-CAM, DeepLIFT) directly on time series without proper adaptation may disadvantage these baselines.  Limited baseline coverage: No comparison with recent time series-specific XAI methods like LIME-TS or kernel-based approaches.
2. Missing Theoretical Guarantees : No completeness axiom (unlike Integrated Gradients), No efficiency property (unlike SHAP). More importantly, no monotonicity - Higher attribution ≠ more important. why and how high scores mean important segments

**Questions:**

1. P loses temporal ordering information. How does this affect attribution for time-dependent patterns?
2. The gradient-based attribution assumes differentiability, how does it interact with ReLU operations.

---

> ### Author Response · Authors · 2025-11-21
> **Response to Reviewer VCe7 [1/4]**
>
> Thank you for your time and constructive feedback. We appreciate your recognition of our method's novelty and intuition and especially our efforts to provide practical and theoretical insights to our design choices. Thank you for recognizing our comprehensive experiments design for evaluations. Below, we address your questions and comments.
>
> ## Weaknesses
>
> ---
> > W1. baseline comparisons: Using computer vision methods (Grad-CAM, DeepLIFT) directly on time series without proper adaptation may disadvantage these baselines. Limited baseline coverage: No comparison with recent time series-specific XAI methods like LIME-TS or kernel-based approaches.
>
> We included methods such as DeepLIFT, Integrated Gradient (IG), and Grad-CAM, which were originally developed for computer vision, because they have been widely adopted as baselines in prior time-series interpretability work with comparable performance [1,2,3,4]. Our evaluation also includes several methods specifically designed for sequential data, such as Attention, Grad-SAM (Attention×Gradient), TimeX++, and LIMESegment (added following reviewer feedback).
>
> In response to the reviewer's comment, we have now incorporated **three new kernel-based approaches as additional baselines: LIME [5], KernelSHAP [6], and LIMESegment [7] (an extension of LIME for time-series data)**. Note that LIMESegment has inherent limitations: it can only be applied to univariate data with shorter sequence lengths (<500). Therefore, we report LIMESegment results only for datasets satisfying these constraints. The comparison with these new baselines is shown in the tables below. **TimeSliver maintains superior explainability performance across all baselines.**
>
> These changes have been incorporated (highlighted in blue) in **Tables 1, 2, 3 and Section 3** of the revised paper.
>
> **Comparison on four synthetic dataset:**
> | Method        | FreqSum        | SeqComb-UV      | SeqComb-MV      | LOWVAR         |
> |---------------|----------------|------------------|------------------|----------------|
> | **LIME**          | 0.36±0.09      | 0.26±0.07        | 0.23±0.07        | 0.10±0.06      |
> | **LIMESegment**   | NA             | 0.76±0.08        | NA               | NA             |
> | **KernelSHAP**    | 0.37±0.06      | 0.30±0.05        | 0.30±0.05        | 0.12±0.05      |
> | **TimeSliver**    | **0.94±0.05**      | **0.97±0.03**        | **0.94±0.01**        | **0.99±0.04**      |
>
> **Comparison of positive attribution score on real-world dataset:**
>
> | Method | Audio I(100)↑ | Audio I(20)↑ | EEG I(100)↑ | EEG I(20)↑ | FORD-A I(100)↑ | FORD-A I(20)↑ |
> |---|---|---|---|---|---|---|
> | LIME | 69.09±0.44 | 9.54±0.37 | 64.02±1.3 | 10.55±0.33 | 66.92±0.46 | 10.17±0.04 |
> | LIMESegment | NA | NA | NA | NA | 76.14±0.53 | 10.57±0.11 |
> | KernelSHAP | 72.20±1.71 | 10.52±0.51 | 66.92±0.46 | 10.47±0.04 | 67.80±1.4 | 11.10±0.30 |
> | TimeSliver | **74.30±0.68** | **11.35±0.15** | **83.99±0.61** | **14.52±0.15** | **93.87±1.01** | **14.99±0.01** |
>
> **Comparison of negative attribution score on real-world dataset:**
> | Method | EEG 2% Masking↑ | EEG 5% Masking↑ | Audio 2% Masking↑ | Audio 5% Masking↑ | FordA 2% Masking↑ | FordA 5% Masking↑ |
> |---|---|---|---|---|---|---|
> | LIME | 1.10±0.03 | 1.15±0.04 | **1.02±0.04** | 0.99±0.03 | 1.01±0.00 | 1.01±0.02 |
> | LIMESegment | NA | NA | NA | NA | 1.01±0.01 | 1.02±0.01 |
> | KernelShap | 1.18±0.01 | 1.24±0.02 | 0.98±0.02 | 0.96±0.01 | 1.02±0.01 | 1.02±0.02 |
> | TimeSliver | **1.31±0.00** | **1.36±0.01** | **1.02±0.04** | **1.01±0.01** | **1.08±0.00** | **1.09±0.00** |

---

> ### Author Response · Authors · 2025-11-21
> **Response to Reviewer VCe7 [2/4]**
>
> > W2.1. Missing Theoretical Guarantees : No completeness axiom (unlike Integrated Gradients), No efficiency property (unlike SHAP).
>
> In the literature, theoretical guarantees for completeness axiom, sensitivity axiom, efficiency and symmetry property are typically proved for post-hoc attribution methods such as IG, DeepLIFT, and SHAP, as they can be applied to any trained/frozen models. However, the models specifically developed for time series data, such as TimeX++ [4], calculate attribution scores by training a non-linear explainer. Thus, the above mentioned properties become intractable for methods with non-linear explainers. However, in the case of $\texttt{TimeSliver}$, the model linearly captures the interaction between temporal segments and converts representation $\boldsymbol{P}$ in Equation 2 to class logits; thus satisfying the properties discussed above under certain conditions.
>
> Below we briefly discuss $\texttt{TimeSliver's}$ theoretically desirable properties (**Completeness, Sensitivity and Symmetry-preserving**)and have added this discussion to **Appendix F** in the revision.
>
> ### Completeness
>
> To satisfy the *completeness* axiom, the attribution scores of all temporal segments for class $c$ in input $\mathbf{x}$ must sum to the change in the predicted logit for class $c$ between $\mathbf{x}$ and a baseline $\mathbf{x}\_{\text{baseline}}$:
>
> $$\hat{y}\_c(\mathbf{x}) - \hat{y}\_c(\mathbf{x}\_{\text{baseline}}) = \sum\_{k=0}^{\kappa-1} \phi\_k,$$
>
> where $\kappa$ is the number of temporal segments and $\phi$ is the attribution score. In this study, we set $\mathbf{x}\_{\text{baseline}} = 0$, so that $\hat{y}\_c(\mathbf{x}\_{\text{baseline}}) = 0$. Hence, we need to show that
>
> $$\hat{y}\_c(\mathbf{x}) = \sum\_{k=0}^{\kappa-1} \phi\_k.$$
>
> As noted in Section 2, $\texttt{TimeSliver}$ transforms $\boldsymbol{P}$ into class logits using just a linear layer. Thus the predicted logit of class $c$ for input $\mathbf{x}$ can be expressed as:
>
> $$\hat{y}_c = \sum\_{i=0}^{n\nu - 1} \sum\_{j=0}^{q-1} w\_{ij}P\_{ij}$$
>
> This implies that
>
> $$\frac{\partial \hat{y}_c}{\partial P\_{ij}} = w\_{ij}$$
>
> Let's assume that $\mathbf{x}$ only consists of positively attributing temporal segments for class $c$. This implies that:
>
> - $\sigma\_{ij} = \text{sign}(w\_{ij}) = +1$ for all $(i,j) \in \{0,\dots,n\nu - 1\} \times \{0,\dots,q-1\}$
> - $Z\_{ki}Q\_{kj} \geq 0$ for all $(i,j,k) \in \{0,\dots,n\nu - 1\} \times \{0,\dots,q-1\} \times \{0,\dots,\kappa-1\}$ in Equation 3
>
> Additionally, based on the empirical results presented in Table 13, we can remove max-scaling in Equation 3 by accepting a decrease in AUPRC score by 1.5%. Based on the above conditions, we can rewrite Equation 3 (in the paper) as:
>
> $$\zeta\_{k,ij}^+(w\_{ij}) = w\_{ij} \times Z\_{ki} Q\_{kj} \quad \text{and} \quad \zeta\_{k,ij}^-(w\_{ij}) = 0$$
>
> Further, the total attribution score of a $k^{\text{th}}$ temporal segment can be calculated as:
>
> $$\phi\_{k}^+ = \sum\_{i=0}^{n\nu - 1} \sum\_{j=0}^{q-1} w\_{ij} \times Z\_{ki} Q\_{kj}$$
>
> Adding the attribution scores of all the temporal segments and using Equation 2 (in the paper) leads to:
>
> $$\sum\_{k=0}^{\kappa-1} \phi\_{k}^+ = \sum\_{i=0}^{n\nu - 1} \sum\_{j=0}^{q-1} w\_{ij} \times \sum\_{k=0}^{\kappa-1} Z\_{ki} Q\_{kj} = \sum\_{i=0}^{n\nu - 1} \sum\_{j=0}^{q-1} w\_{ij} P\_{ij} = \hat{y}_c$$
>
> The above equations demonstrate that $\texttt{TimeSliver}$ satisfies the *completeness* axiom when all temporal segments have positive attributions for class $c$. It is worth noting that the axiom is also satisfied when all segments have negative attributions for class $c$ in $\mathbf{x}$.

---

> ### Author Response · Authors · 2025-11-21
> **Response to Reviewer VCe7 [3/4]**
>
> ### Sensitivity
>
> To satisfy the *sensitivity* axiom, any change or perturbation ($\delta\_s$) to a temporal segment $\mathbf{x}\_s = \mathbf{x}[t:t+m]$ that induces a change in the predicted logit $\hat{y}\_c$ must yield a nonzero attribution score for that segment, i.e.,
>
> $$\hat{y}\_c(\mathbf{x}) \neq \hat{y}\_c(\mathbf{x}\_{\{x\_s \leftarrow x\_s + \delta\_s\}}) \;\implies\; \phi\_s \neq 0,$$
>
> In $\texttt{TimeSliver}$, the attribution score computation (Equations 3 and 4) involves two key steps after training:
>
> - **Sensitivity of $\hat{y}\_c$ with respect to $\boldsymbol{P}$.**
>   We first estimate $g\_{ij} = \frac{\partial \hat{y}\_c}{\partial P\_{ij}}.$ Although gradient-based methods typically violate sensitivity [8], $\texttt{TimeSliver}$ preserves it because $\hat{y}\_c$ depends *linearly* on $\boldsymbol{P}$ (Equation 9). Thus, any perturbation in $\boldsymbol{P}$ that affects $\hat{y}\_c$ necessarily yields $g\_{ij} \neq 0$.
>
> - **Separating positive and negative contributions.**
>   In the second step, we use $\sigma\_{ij} = \operatorname{sign}(w\_{ij})$ together with the $Z\_{ki} Q\_{kj}$ terms in Equation 3 to isolate positively and negatively contributing segments via the $\operatorname{ReLU}$ operator. Consequently, $\texttt{TimeSliver}$ satisfies sensitivity *partially*: $\phi^{+}$ is insensitive to negatively contributing segments (they are zeroed), and $\phi^{-}$ is insensitive to positively contributing segments.
>
> ### Symmetry-Preserving
>
> For a method to be symmetry-preserving, it must assign identical attribution scores ($\phi^+$ and $\phi^-$) to two identical temporal segments in the input sequence. To illustrate that TimeSliver satisfies this property, consider a univariate time-series instance $\mathbf{x} \in \mathbb{R}^{L \times 1}$ containing two identical segments $\mathbf{x}_s = \mathbf{x}[t_1:t_1+m]$ and $\mathbf{x}'_s = \mathbf{x}[t_2:t_2+m]$. As shown in Equation 3, the attribution score for a segment $\mathbf{x}_i$ primarily depends on $Z_i$ and $Q_i$. Section 2.2.2 establishes that $\boldsymbol{Z}$ depends solely on the one-hot encoding $\mathcal{O}$ (Equation 1). Because identical raw segments yield identical one-hot encodings, we have $\mathcal{O}_s = \mathcal{O}'_s$. Likewise, the latent representation $\mathbf{q}_i$ is obtained from a 1D CNN applied at the segment level; therefore, identical segments produce identical latent representations, i.e., $\mathbf{q}_s = \mathbf{q}'_s$. Consequently, both $Z$ and $Q$ are identical for the two segments. By Equation 3, this yields identical attribution scores $\zeta^+$ and $\zeta^-$, and therefore identical final attributions $\phi^+$ and $\phi^-$.
>
> > W2.2. No monotonicity - Higher attribution ≠ more important. why and how high scores mean important segments
>
> In the time-series interpretability literature, attribution score magnitude directly correlates with temporal segment importance [1,2,3,4,5,6]. This fundamental relationship underlies the use of AUPRC as a standard metric for evaluating how effectively explainability methods identify critical temporal segments. TimeSliver's superior AUPRC scores in **Table 1** demonstrate that our method produces attribution scores that are strongly proportional to ground truth importance scores. This quantitative finding is further validated by the qualitative analysis in **Figure 9 (Appendix)**, which shows clear visual alignment between \texttt{TimeSliver}'s attributions and ground truth importance patterns across multiple datasets.
>
> For real-world datasets, where ground truth importance scores are unavailable, we validate this alignment through masking experiments (**Figure 8, Appendix**). TimeSliver achieves the highest accuracy when retaining only the top 5--10\% of time points ranked by attribution scores, demonstrating that our method successfully identifies the most predictive temporal segments for the model's decisions. Additionally, we have updated **Definition 2.1** to explicitly describe positive and negative attribution scores in the context of our work, in line with previous works for clarity.

---

> ### Author Response · Authors · 2025-11-21
> **Response to Reviewer VCe7 [4/4]**
>
> ## Questions:
> > Q1. P loses temporal ordering information. How does this affect attribution for time-dependent patterns?
>
> It is correct that the temporal ordering is lost in the construction of $\boldsymbol{P}$ and we have taken this into consideration in our initial design by including positional encoding in **Module I** based on the datasets. For datasets where the discriminative features are more localized (e.g., localized frequency patterns), $\boldsymbol{Z}$ and $\boldsymbol{Q}$ are sufficient for effective classification. However, for tasks requiring explicit temporal ordering to capture sequential dependencies, we add sinusoidal positional encodings [9] to $\mathbf{x}_i$ in **Module I (Section 2.2.1)** to calculate $\boldsymbol{Q}$, while $\boldsymbol{Z}$ remains unaffected. The inclusion of positional encoding for each dataset is determined empirically by comparing predictive performance with and without it.
>
> We have explicitly added this description in **Section 2.2.3** to highlight the use of positional encoding and have expanded **Appendix B.3** to include more implementation details.
>
> > Q2. The gradient-based attribution assumes differentiability, how does it interact with ReLU operations.
>
> We want to clarify that the gradient computation and ReLU operations occur at distinct stages, ensuring full differentiability throughout the attribution process.
>
> The computation of positive ($\phi^+$) and negative ($\phi^-$) attribution scores using Equations 3 and 4 relies on a single gradient term $g\_{ij}=\frac{\partial \hat{y}\_c}{\partial P\_{ij}}$. As established in Section 2.2 and Equation 9, the mapping from $\boldsymbol{P}$ to the predicted logit $\hat{y}\_c$ is strictly linear:
>
> $$\hat{y}\_c = \sum\_{i=0}^{n\cdot\nu - 1} \sum\_{j=0}^{q-1} w\_{ij}P\_{ij}$$
>
> **This linear relationship is maintained by design**: the final classification layer contains no nonlinear activations. Consequently, the gradient $g\_{ij} = w\_{ij}$ is well-defined and differentiable everywhere.
>
> **The ReLU operation in Equation 3 serves a different purpose**: it operates *post-gradient* to decompose the linear contributions $Z\_{ki}Q\_{kj}$ into directionally signed components (positive vs. negative attributions). Importantly, ReLU is **not** part of the forward prediction path from $\boldsymbol{P}$ to $\hat{y}\_c$; it is applied only during the attribution calculation to separate $\phi^+$ and $\phi^-$. Therefore, the gradient-based attribution computation remains fully differentiable, and the use of ReLU does not introduce any gradient flow issues.
>
> [1]  Evaluation of post-hoc interpretability methods in time-series classification, Nature Machine Intelligence, 2023.
>
>
> [2]  Interpretable Convolutional Neural Network through Layer-wise Relevance Propagation for Machine Fault Diagnosis, IEEE Sensor, 2019.
>
>
> [3] Inherently Interpretable Time Series Classification via Multiple Instance Learning, ICLR, 2024
>
>
> [4] TIMEX++: LearningTime-SeriesExplanationswithInformationBottleneck, ICML, 2024.
>
>
> [5] “Why Should I Trust You?” Explaining the Predictions of Any Classifier, KDD, 2016.
>
>
> [6] A Unified Approach to Interpreting Model Predictions, NIPS, 2017.
>
>
> [7] LIMESegment: Meaningful, Realistic Time Series Explanations, AISTATS, 2022.
>
>
> [8]  Learning important features through propagating activation differences, ICML, 2017.
>
>
> [9] Attention Is All You Need, NIPS, 2017.

---

> > ### Comment · Reviewer_VCe7 · 2025-11-28
> >
> > Thanks for the authors' response. I will raise my ratings from 4 to 6.

---

> > > ### Author Response · Authors · 2025-11-28
> > >
> > > We are glad that our responses clarified your questions. Thank you for raising our score based on our response. We wanted to kindly mention that the updated score has not yet been reflected in the OpenReview portal.

---

### Author Response · Authors · 2025-12-03
**Global Response to ACs/SACs [1/2]**

Dear AC,

Thank you for your time and effort in overseeing our paper. We are grateful for the reviewers' feedback and discussion. We are writing to provide a summary of our revisions and discussion with the reviewers, which has yielded **a unanimous positive rating for our paper**. We summarize the four key aspects below: 1) our strengths and contributions recognized by the reviewers , 2) the additional experiments we added, 3) the major clarifications we provided to the reviewers, and 4) the summary of our discussion with the reviewers.

## 1. Strengths and Contributions

**Novel architecture**: Reviewers **VCe7** and **5bwY** found the combination of symbolic abstraction with raw time series through linear decomposition to be novel and well-motivated. Reviewer **VCe7** additionally welcomed the analogy of symbolic abstraction with STFT.

**Strong evaluation methods and experimental results**: Reviewers **VCe7**, **A8un**, and **5bwY** found the experimental results to be thorough and indicative of $\texttt{TimeSliver}$'s strong performance in estimating temporal attribution scores. Reviewer **Cxgd** found the evaluation strategy in the paper to be very sound.

**Positive and Negative attribution scores**: Reviewer **5bwY** found the division of positive and negative attribution scores to be a valuable contribution, as it provides insight into the directionality of impact on the model's output.

-----

## 2. Additional Experiments

**Added 3 more baselines (Table 1, 2, and 3)**: Based on Reviewer **VCe7**'s comment, we included kernel-based explainability methods (LIME, KernelSHAP, and LIMESegment) as our baselines and updated Tables 1, 2, and 3 in the paper. $\texttt{TimeSliver}$ outperforms the newly added baselines as well.

**Updated the evaluation strategy for Negative temporal attribution scores (Section 2.2.5 and Table 3)**: Based on Reviewer **Cxgd**'s comment, we changed our evaluation strategy for negative attribution scores. To verify whether the model correctly identifies negatively attributing time points, it is more precise to study the change in the logit value of the predicted class after masking the negatively attributing time points. We updated Table 3 by reporting the change in the logit of the predicted class rather than the change in overall accuracy after masking negatively attributing time points (2% and 5% masking). $\texttt{TimeSliver}$ maintains its position as the top performer with the new evaluation strategy as well. After we updated the evaluation strategy for negative attribution scores, the reviewer was satisfied and raised the score.

**Ablation study to showcase the effect of different activation functions on the attribution scores (Table 13 in Appndix)**: Based on Reviewer **5bwY**'s comment, we included an ablation study in Table 13 to show the impact of different activation functions (Sigmoid, Tanh) in Equation 3. The ablation study confirms that ReLU is the best choice for attribution score calculation.

**Effect of different symbolic representation of predictability (Figure 3 and Table 9)**: Based on Reviewer **A8un**'s comment, we revised Figure 3 (Section 2.2.4) and Table 9 (Appendix D.2) to illustrate the effect of replacing symbolic discretization $Z$ (Module II, Section 2.2.2) with non-symbolic projection $X_{proj}$ and choosing other symbolic representations (ABBA, SFA) on the predictive performance. The results clearly show that the choice of different symbolic representations (or the absence of them) has minimal impact on predictability.

**$\texttt{TimeSliver}$'s theoretically desirable properties (Appendix F)**: Based on Reviewer **VCe7**'s comment, we added an additional section (Appendix F) that mathematically discusses the theoretically desirable properties (Completeness, Sensitivity, and Symmetry-Preserving) of $\texttt{TimeSliver}$.

---

> ### Author Response · Authors · 2025-12-03
> **Global Response to ACs/SACs [2/2]**
>
> ## 3. Major Clarifications for reviewers
>
> **Definition of positive and negative attribution**: Based on comments from Reviewers **Cxgd** and **5bwy**, we included the definition for positive and negative attribution scores in Definition 2.1.
>
> **Step to capture time-dependent patterns**: Although the temporal ordering is lost through the construction of $\boldsymbol{P}$ in Equation 2, the positional information is incorporated into the model by adding positional encoding to the one-hot encoding before it enters the 1D CNN network in Module I. To clarify this point raised by Reviewer **VCe7**, we added a few lines in Module III to clearly highlight this aspect.
>
> **Gradient term used in Equation 3 is not impacted by ReLU's discontinuous gradient**: The computation of positive and negative attribution scores using Equations 3 and 4 relies on a single gradient term $g_{ij}$ . As established in Section 2.2 and Equation 9, the mapping from $\boldsymbol{P}$ to the predicted logit $\hat{y}\_c$ is strictly linear; thus, the gradient $g_{ij}$ is well-defined and differentiable everywhere.
>
> **$\texttt{TimeSliver}$ captures higher-order interactions**: We clarified in our response to Reviewer **5bwY** that through the combination of non-linear latent representation (Module I, Section 2.2.1), positional encoding (Module III, Section 2.2.3), and multi-scale temporal modeling (Module III, Section 2.2.3), $\texttt{TimeSliver}$ captures higher-order interactions in time-series data. To demonstrate this further, we showcased $\texttt{TimeSliver}$'s ability to capture temporally disjoint yet jointly informative patterns through a synthetic dataset in Appendix D.3.
>
> ----
>
> ## 4. Reviewers Response
>
> **Reviewer Cxgd**: After we rewrote Definition 2.1 to clarify positive and negative attribution scores and updated our notations (mainly in Section 2.1) to clarify the problem statement, and also updated the evaluation strategy for negative attribution scores, the **reviewer was satisfied** with all the changes and **increased the overall score along with the individual subscores for soundness and presentation**.
>
>
> **Reviewer VCe7**: After we included 3 additional baselines (LIME, KernelSHAP, LIMESegment), clarified the theoretical properties of $\texttt{TimeSliver}$, and explained the inclusion of positional encoding for capturing time-dependent patterns, the **reviewer was satisfied with our response and raised the score**.
>
> **Reviewer 5bwY**: The reviewer was satisfied with all the changes made in response to the questions raised and decided to **maintain the positive rating**.

---

### Meta-Review · Area_Chair_hNT6 · 2026-01-04

**Summary:**

The reviewers main concerns were missing baselines and multiple clarity issues. The rebuttal addresses these points directly with new experiments and writing, so the strengths outweigh the weaknesses if the changes are made for the next version of the paper. Especially to highlight comparisons against time series-specific baselines, of which there are now many.

**Reviewer Concerns:**

Please see the overview above.

**Reviewer Scores:**

All reviewers had a chance to improve their scores.

---

### Decision · Program_Chairs · 2026-01-26

Accept (Poster)